# Taxonomy, Opportunities, and Challenges of Representation Engineering for Large Language Models

**Jan Wehner**[1]*,   **Sahar Abdelnabi**[2],   **Daniel Tan**[3],   **David Krueger**[4],   **Mario Fritz**[1]

[1] *CISPA Helmholtz Center for Information Security*
[2] *Microsoft*
[3] *University College London*
[4] *Mila, University of Montreal*

**Reviewed on OpenReview:** `https://openreview.net/forum?id=2U1KIfmaU9`

## Abstract

Representation Engineering (RepE) is a novel paradigm for controlling the behavior of LLMs. Unlike traditional approaches that modify inputs or fine-tune the model, RepE directly manipulates the model's internal representations. As a result, it may offer more effective, interpretable, data-efficient, and flexible control over models' behavior. We present the first comprehensive survey of RepE for LLMs, reviewing the rapidly growing literature to address key questions: What RepE methods exist and how do they differ? For what concepts and problems has RepE been applied? What are the strengths and weaknesses of RepE compared to other methods? To answer these, we propose a unified framework describing RepE as a pipeline comprising representation identification, operationalization, and control. We posit that while RepE methods offer significant potential, challenges remain, including managing multiple concepts, ensuring reliability, and preserving models' performance. Towards improving RepE, we identify opportunities for experimental and methodological improvements and construct a guide for best practices.

## 1 Introduction

Prompting and fine-tuning are common and effective methods for controlling the behavior of Large Language Models (LLMs). But recently, a new paradigm for controlling LLMs inspired by interpretability research has emerged: Representation Engineering (RepE). Instead of adapting the inputs or training the weights towards outputs, Representation Engineering controls the LLMs' behavior by manipulating its internal representations. For this, it first identifies how a human-understandable concept is represented in the network's activations. Next, it uses that knowledge to steer the model's representations, thus, controlling its behavior (see Figure 1).

By tapping into models' representations, RepE offers two main advantages: (1) **Improve understanding**: RepE identifies how human-understandable concepts are represented in the models' activation space; steering that representation can verify whether the representation has the expected influence on the outputs. (2) **Control**: RepE is promising as a powerful tool to control the behavior, personality, encoded beliefs, and performance of an LLM, allowing us to make models behave in safe and desired ways. Since no training is required, RepE can be cheaper, more data efficient, and more flexible to different users and situations than other methods while causing less deterioration to the model's performance.

Initial work on Activation Steering (Turner et al., 2024; Li et al., 2023a) was built on the assumption that concepts are represented as linear directions in the activation space of LLMs (Park et al., 2024b). These

---

*Correspondence to jan.wehner@cispa.de

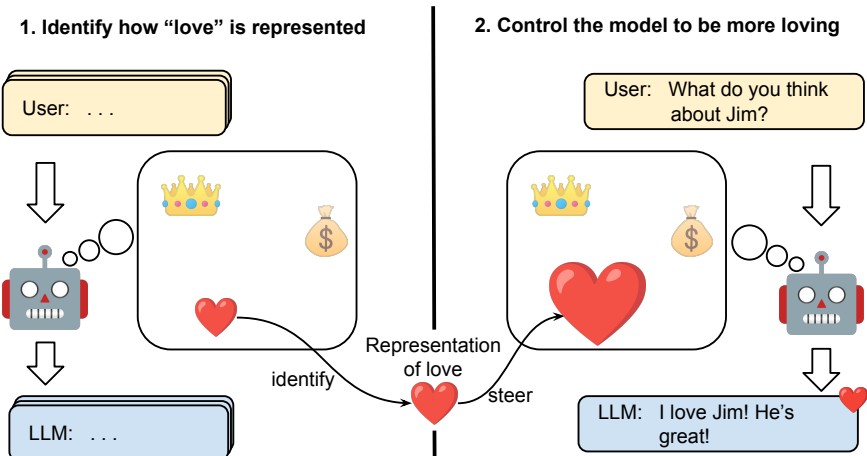

Figure 1: Representation Engineering first **identifies** how a concept is represented in the activation space of the model and then steers that representation to **control** the model's behavior.

methods focused on the difference in activations for inputs that are positive or negative with regards to the concept. They identify a vector that captures the model's representation of a concept and can be added to the activations on new inputs to modulate the intensity of the concept. Since then, a range of new RepE methods have been proposed. For example, such methods identify representations by finding interventions that lead to desired outputs (Cao et al., 2024a) or learning internal features in an unsupervised fashion (Templeton et al., 2024). They go beyond static and linear representations (Qiu et al., 2024) by employing different operators than a vector (Postmus & Abreu, 2024). And they experiment with new functions for steering activations (Singh et al., 2024), as well as modifying the weights of the model by adding adapters (Wu et al., 2024c).

However, the literature on RepE lacks an overview and there exist many unanswered questions and conceptual confusions. In our work, we systematically answer the following questions.

- **What is Representation Engineering?** We provide the first unification of different approaches to RepE with a framework that describes them as pipelines that identify, operationalize, and control representations.
- **What RepE methods exist and how do they differ?** We discuss and contrast different methods for each step in this pipeline.
- **How are RepE methods evaluated?** We describe evaluation methodologies and benchmarks, and propose best practices for evaluating RepE methods.
- **How has RepE been applied?** We describe how RepE has been applied to areas such as AI Safety, Ethics, and Interpretability and show which concepts are more amendable to be controlled by RepE.
- **How does it compare to other methods?** We contrast RepE to related methods and provide a meta-survey of comparisons between RepE and fine-tuning, prompting, and decoding-based methods.
- **Why does it work?** We suggest reasons why RepE works so effectively.
- **What are strengths and weaknesses of RepE?** We provide a list of advantages and challenges of RepE.
- **What opportunities for future research are there?** We suggest broad themes and concrete ideas for improvements and outline opportunities for strengthening the field.

To answer these questions we conduct a thorough literature review, collecting detailed information from >130 papers. Although previous surveys on Mechanistic Interpretability (Ferrando et al., 2024; Bereska & Gavves, 2024) and frameworks for causal interpretability (Mueller et al., 2024; Geiger et al., 2024a) have touched on RepE, there is no dedicated survey of this research area. A survey is especially pressing since over 100 papers have been released within the last year without a systematization of the literature. In addition to

Activation Steering, we include methods that modify the weights, identify concept representations through optimizing for outputs, and are not based on vectors. We focus on methods that apply control on intermediate representations of a model without fully replacing them, thus excluding soft-prompting, decoding-based methods, and Activation Patching. For a detailed description of the inclusion criteria and literature search process, see Appendix B.

We conclude that future work needs to build ways to benchmark RepE methods to enable thorough comparisons. In addition, we identify opportunities in considering representations that are non-linear, multi-concept, have interactions between layers or trajectories over time. Furthermore, there are a number of promising directions for new applications, and improved concept identification and control methods. To make RepE usable in practice, we will need to see improvements in its ability for multi-concept steering, long-form generation, reliability, Out-of-Distribution (OOD) robustness, and maintenance of general capabilities.

## Contents

## 2 What is Representation Engineering?

Representation Engineering (RepE) is a class of techniques that: **Manipulate representations of a model in order to control its behavior with regard to a concept**.

One perspective on RepE frames the computations in an LLMs as a computer program with intermediate variables and computations that use these variables. In this perspective we can attempt to find intermediate variable that correspond to specific concepts. This allows us to meaningfully change the values of these variables to influence later computations and ultimately the outputs of the program. Another perspective states that RepE aims to identify patterns in the activations of an LLM that correspond to high-level, human-understandable concepts. It turns out that these activation patterns can be used to manipulate the activations to get consistent changes in the behavior of the LLM.

**Goals of RepE.** The goals of RepE are (1) Behavior Steering: Our ability to control the behavior of LLMs and (2) Interpretability: Our ability to understand the internal computations of LLMs. RepE can be used by LLM providers to prevent undesired behavior of an LLM such as guarding against jailbreaks. It can also be used to finely control the behavioral tendencies of the LLM or to improve it's suitability and performance at specific tasks for example by adapting it's personality and style or by improving it's reasoning performance. Additionally, RepE can provide us with some insight into the internal processes of an LLM which are commonly considered as a black-box. By demonstrating that some concepts are consistently represented in LLMs, determining in how much a concept's representation is active on specific outputs or showcasing the influence that a concept has on the output of the model. As such RepE is often a tool for scientific inquiry into specific LLM behaviors.

**Running Example.** One of the most commonly steered concepts is Truthfulness, a models intention to say things that are true instead of false as for example measured in the TruthfulQA benchmark. RepE methods aim to identify patterns in the activations that correspond to the models representation of the concept. This could for example be a linear direction whose magnitude denotes how truthful the outputs of an LLM are on a specific inputs. This activation pattern, hereforth called concept operator, can then be used to detect when the model is being dishonest and to steer it to give truthful outputs. As such it could be deployed by model providers to decrease hallucinations and be used by safety researchers to detect and mitigate cases of deceptive behavior.

**The RepE Pipeline.** RepE methods consist of a pipeline with two steps: First, we identify how the targeted concept is represented within the model. Secondly, we use that information to steer the model's representations on new inputs. In this context, a concept can be understood broadly as any human-understandable

feature like behaviors, tasks, entities, pieces of knowledge, or personality traits. RepE assumes that such concepts are represented in LLMs. Representations are the internal structures or encodings that a model uses to capture and process information about a concept.

The goal of **Representation Identification** is to produce a **concept operator** that accurately captures the model's representation of a concept. The concept operator is an object, for example, a vector, with a specific intended relationship to the model's actual concept representation. This bakes in assumptions about the geometry and location of concept representations.

The concept operator is then used for **Representation Control** by steering the representations of the model. For this a **steering function** is employed that uses the concept operator to manipulate the activations or weights of the model. If the concept operator accurately reflects the model's representations and the steering function is effective, this will steer the model's representations and, thus, control the outputs of the model.

RepE is also sometimes referred to as Activation Engineering or Activation Steering. However, we chose the term Representation Engineering to emphasize that representations can also be controlled by modifying the weights of a model.

## 2.1 Strategies for RepE

There are a variety of techniques to perform Representation Identification and Control. The most common way to perform RepE is through Linear Activation Addition. Hereby, a linear direction is identified that should denote the representation of a concept for example by contrasting activations for inputs where the concept is or isn't present. Another popular approach are Sparse Autoencoders, that disentangle features into a dictionary of concepts by projecting them into a higher-dimensional, sparse space. Finally, it's possible to directly identify representations that cause the specific behavior changes by fine-tuning a concept operator towards desired outputs.

Once the representation was identified, we can use it to control the LLMs behavior by adapting it's activations or changing it's weights. For example we could simply add a vector on the activations for new inputs. Or we can adapt the weights, such that they produce activations that align more strongly with the activation patterns we have identified.

**Illustrative Examples** To illustrate RepE, we provide an example inspired by Zou et al. (2023a), where RepE is applied to make an LLM give more truthful answers. To identify how truthfulness is represented, inputs are provided that instruct the model to be honest or to lie. Next, the activations for these inputs are collected. Then, the difference between activations for honest and dishonest inputs is calculated. This produces a vector that denotes a direction in the activation space which represents honesty. This vector can be used to guide the training of a LoRA module that generates activations more aligned with the truthfulness direction. This effectively steers the model to give more truthful answers.

In another example inspired by Cao et al. (2024a), RepE is applied to align the outputs of an LLM with the preferences of a human. For this, they find how activations need to be changed to lead to desired outputs. Given a dataset of desired and undesired outputs, a vector is optimized that can be added to the activations during a forward pass to make it more likely for the desired output to be generated. When processing new inputs, that vector can be added to the activations, thus, steering the model towards human preferences.

## 2.2 Advantages of Representation Engineering

**Sample Efficient.** RepE can be effective with a low number of training examples (Wang et al., 2024a). Furthermore, some RepE methods do not require a labeled dataset (see Section 7.1). These properties make it easier and cheaper to employ RepE.

**Flexible.** The steering of RepE can be turned on or off flexibly. It is also possible to dynamically adjust the steering per request, thus enabling personalization and context-dependent control (Cao et al., 2024a; Stickland et al., 2024; Guo et al., 2024; Chen et al., 2024a; Lucchetti & Guha, 2024).

**Low Impact on Capabilities.** As shown in Section 11.2, RepE tends to not strongly degrade the capabilities of the model (Panickssery et al., 2024; Stickland et al., 2024; Guo et al., 2024; Qiu et al., 2024; van der Weij et al., 2024). Thus, it is possible to use RepE practically in production without sacrificing model quality.

**Precise Control.** RepE offers direct control over the representations of a concept. This makes it possible to apply precise steering at the granularity of single concepts.

**Efficient During Inference.** Methods for Representation Control come at a negligible increase in computational cost during inference (Li et al., 2023a). Most methods do not require a large number of additional parameters, and none require additional inference steps.

**Causally Verifiable Interpretability.** By observing the effect of steering a representation on the output, we can verify that the identified representation causes the expected change (Marks & Tegmark, 2023; Arora et al., 2024). This can make us more confident that the right representation for a concept was found.

**Tacit Interpretability.** By giving users access to knobs they can turn to influence internal, human-understandable representations, RepE can provide tacit understanding to non-expert users about the inner workings of LLMs (Chen et al., 2024a).

**Advantages over Other Methods.** Our meta-survey in Section 11.2 indicates that RepE tends to be more effective at controlling models' behavior than alternative methods like prompting, fine-tuning, or decoding-based methods.

**Compatible with Other Methods.** RepE can be used in addition to other methods for stronger control (see Section 11.2). This enables the use of RepE in a defense-in-depth approach, where multiple imperfect methods are combined to achieve strong safety guards.

> **Takeaway: What is Representation Engineering?**
> RepE manipulates the representations of a model in order to control its behavior w.r.t. a concept. It can be used to steer a models behavior and interpret its internals. This is done by identifying how a concept is represented within the model and then using that knowledge to steer the representations. It allows for effective, efficient, flexible, and precise control over models' behavior while largely retaining quality.

## 3 Framework and Notation

Building on the background and definitions in Section 2, we propose a framework that classifies RepE pipelines based on the method they use for Representation Identification, Representation Control and how they operationalize representations (see Figure 2). Additionally, we provide a unifying formalization of RepE that showcases the function signatures of the steps of RepE pipelines. For an overview of the notation, refer to Table 1.

### 3.1 Representation Identification

Representation Identification (RI) is the first step in the RepE pipeline. It aims to find out how the concept is represented in the LLM's activations. For this, an RI method needs to provide a stimulus related to the concept, collect the model's activations, and calculate the representation of the concept from the collected activations. The result is a concept operator that denotes the model's representation of the concept.

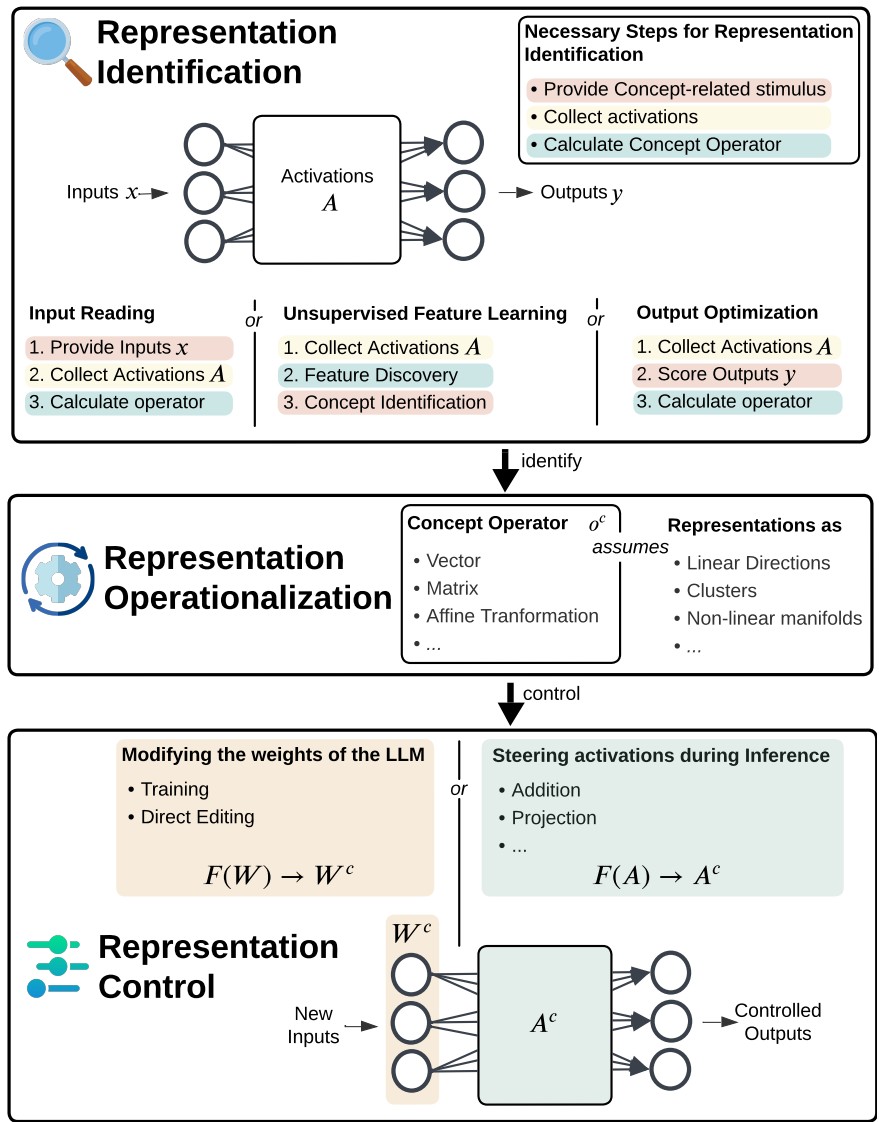

Figure 2: Framework of Representation Engineering pipelines. One Representation Identification method is used to identify a concept operator. Representations are operationalized by assuming a geometry of representations. The concept operator is used to steer the weights or activations of the model.

**Definition 1.** *Representation Identification RI is a method*

$$RI(M, S) \rightarrow o^c$$

*that takes*

- *a model $\mathcal{M}$*

- *a set of strings $S$ that have a positive, negative, neutral, or an unknown relationship to the concept: $S = S^{c^+} \cup S^{c^-} \cup S^{c^\varnothing} \cup S^{c^?}$,*

*and returns a concept operator $o^c$ that denotes the concept's representation.*

Methods for Representation Identification need to carry out three steps. First, they need to *provide a stimulus* related to the concept. These are strings used as inputs or ground-truth outputs that have a

Table 1: Explanation of Notation for ease. Large letters $C, S, X, Y, A, O$ refer to a set of the corresponding elements.

| Notation | Name | Explanation |
|---|---|---|
| $c$ | Target concept | The concept we want to control |
| $\mathcal{M}$ | Model | The LLM on which RepE is applied |
| $s$ | String/set of strings | The text or set of texts |
| $x \in S$ | Input | String given as inputs to the model |
| $y \in S$ | Output | String outputted by the model |
| $s^{c+/-/0}$ | Positive/negative/neutral string | Text that is positively/negatively/neutrally related to the target concept |
| $w_l$ | Weights | Weights of model $M$ at layer $l$ |
| $a_l, m, p$ | Activations | Activations at layer $l$, model component $m$, and token position $p$ |
| RI | Representation Identification | Methods that output the concept operator |
| RC | Representation Control | Methods that steer the model's representations |
| $o^c$ | Concept operator | An object denoting the model's representation of the concept |
| $F$ | Steering function | A function that modifies the weights or activations using $O^c$ |

positive, negative, neutral, or unknown relationship to the concept. This means that they exemplify or contain the concept, serve as counterexamples to the concept, are unrelated to the concept, or that their relationship to the concept is unknown. In our running example these strings might be truthful or deceptive outputs, text that is neither honest nor dishonest or we might just not know it's relationship to honesty. This relationship is usually predefined by humans, e.g., as labels in a dataset. However, in some cases, it is evaluated dynamically during the execution of the method. It is important that these strings actually activate the desired concept's representation, so it is present in the activations. Secondly, they *collect activations* of the model related to these strings.

**Definition 2.** *The activation collection function Act*

$$Act(\mathcal{M}, X^c, l, m, p) \rightarrow A_{l,m,p}^c$$

*extracts activations of model $\mathcal{M}$ for inputs $X$ at layer $l$, model component $m$, and token position $p$*

Crucially, our knowledge about the strings' relationship to the concept lets us infer whether the concept representation is present in the activations for that input. Thirdly, they *calculate the concept operator* from these collected activations. This can range in complexity from a simple arithmetic operation to a large training run.

There are three broad categories of approaches to Representation Identification. Methods that employ *Input Reading* provide inputs related to the concept, read out their activations, and then calculate the operator from the resulting sets of activations. Those that do *Output Optimization* utilize a way of scoring outputs with regard to the concept to optimize the activations accordingly. A concept operator can then be derived from the optimized activations. Finally, other methods employ *Unsupervised Feature Learning*, where they first learn internal features from activations in an unsupervised way. We can identify the feature corresponding to the targeted concept by studying inputs and outputs related to that feature.

### 3.2 Operationalizing Representations

To be able to identify and control representations, RepE pipelines need to operationalize how the model represents the concept. This introduces assumptions about the structure and geometry the model uses to represent the concept in the activations. For example, concepts could be represented as linear directions in the activation space (Park et al., 2024b) or as non-linear manifolds. Furthermore, a concept operator $o^c$ is necessary to denote how the model represents the concept. $o^c$ results from the Representation Identification step and is used in Representation Control. It can take various forms and differs in its intended relationship to the model's actual representation of the concept.

### 3.3 Representation Control

Representation Control (RC) is the second step in a RepE pipeline. RC methods should steer the representation of a concept in order to control the model's behavior w.r.t. that concept. For this, they employ a steering function $F$ that uses the concept operator $o^c$ to modify the model's activations or weights. While $F$ only describes a mathematical operation, RC encapsulates the whole process of where, when, and how $F$ is applied. Employing RC should result in controlled outputs $y^c$ that are more aligned with the concept $c$ than an originally uncontrolled output $y^{\mathrm{org}}$.

**Definition 3.** *Representation Control RC*

$$RC(F, o^c, \mathcal{M}, x) \to y^c$$

*applies the steering function $F$ and concept operator $o^c$ to control the representations of model $\mathcal{M}$ on inputs $x$. This leads to outputs $y^c$ that have been controlled with regard to $c$.*

To steer the representations of a model, we can either modify its activations during inference or adapt its weights. Activations of unseen inputs can be adapted during inference by shifting them using the concept operator.

**Definition 4.** *An activation steering function*

$$F^a(a_l, o_l^c) \to a_l^c$$

*adapts the activations $a_l$ at layer $l$, using $o_l^c$ to attain the steered activations $a_l^c$.*

In addition, we can modify weights to steer the activations they output in desired ways.

**Definition 5.** *A weight steering function*

$$F^w(w_l, o_l^c) \to w_l^c$$

*modifies the weights $w_l$ using $o_l^c$ so that*

$$w_l^c(a_{l-1}) \to a_l^c$$

*the modified weights $w_l^c$ produce steered activations $a_l^c$.*

Some steering functions apply simple operations, while others require retraining of the model. Furthermore, some RC methods are adaptive to the present context, while others always apply the same effect. Weight steering functions only need to be applied once, while activation steering functions need to be applied repeatedly on every new input. Notably, the line between activation and weight steering functions can be blurry. For example, in an MLP layer, adding a steering vector to the activations is equivalent to adding it to the bias term.

> **Takeaway: Framework & Notation**
> RepE is a 2-step pipeline:
>
> 1. **Representation Identification** finds a **concept operator** that denotes the model's representation of the concept.
> 2. **Representation Control** applies a **steering function** that uses the concept operator to modify the activations or weights of the model.
>
> Further, RepE pipelines make assumptions about representations to **operationalize** them into a concept operator.

## 4 Representation Identification

Representation Identification (RI) aims to find a concept operator that denotes how the model represents the target concept in its activations. Methods can be based on Input Reading, Output Optimization, or Unsupervised Feature Learning. This section separately dives into these categories and identifies differences of methods within the categories.

### 4.1 Input Reading

RI methods that fall into the *Input Reading* category provide inputs related to the concept, read out their activations, and finally calculate the operator from the resulting sets of activations. In our running example we can feed inputs related to honest or dishonest behavior into the model, collect their activations and identify patterns that correspond to honesty.

Table 2: Steps and differences between different **Input Reading** methods for **Representation Identification**.

| Steps in Input Reading | How do methods differ? |
|---|---|
| 1. Constructing Inputs | **Source** of inputs (existing datasets, manually written, or synthetically generated) 
 **Relationship** to the concept (positive, neutral, unrelated, or unknown) 
 **Differences** between contrastive sets (paired with concept-related difference, or unpaired) 
 **Prompting template** (pre-prompt concept-related instructions, A/B choices) |
| 2. Collecting Activations | **Layer** (all layers, or concept-related layers only) 
 **Model component** (residual stream, output of MLP or attention heads) 
 **Token position** (End of sequence, all tokens, or concept-related tokens) |
| 3. Calculating the Concept Operator | What **calculation**? (class-mean, probes, or components after dimensionality reduction) 
 Other **techniques** (refining via concept-related neurons, project to other spaces) |

#### 4.1.1 Constructing Inputs

Input Reading methods aim to activate the concept representation by providing inputs $S$ that are related to the concept. To do that, one needs to attain a relevant dataset, divide it into groups that differ in their relation to the concept, and put the inputs into a prompting format that successfully activates the concept's representation.

**Source of Inputs.** The data can be gathered from an existing dataset, manually created, or synthetically generated. A majority of papers use existing datasets like TruthfulQA (Lin et al., 2022) or ToxiGen (Hartvigsen et al., 2022). Researchers can use the labels in the dataset to separate examples into different groups. Multiple papers also use manually created datasets (Li et al., 2024b; Marks & Tegmark, 2023), which require human effort but may allow for more flexible and rigorous experimentation. Furthermore, it is also possible to prompt an LLM to synthetically generate the inputs. This is an efficient and flexible approach that allows targeting concepts for which no dataset exists. However, one might introduce confounders based on the LLM's biases and errors.

**Relationship to the Concept.** Typically, inputs are divided into multiple sets that differ in their relation to the concept. Inputs can be positive $s^{c^+}$, negative $s^{c^-}$, or neutral $s^{c^\emptyset}$ if they exemplify or contain the concept, serve as a counterexample, or are unrelated to the concept, respectively. Inputs can be classified based on whether they mention a certain topic, e.g., by checking for wedding-related keywords (Turner et al., 2024) or containing a specific word (Makelo et al., 2025). More abstractly, inputs can differ in whether they exemplify a concept, e.g., being written in a specific style (Konen et al., 2024). Lastly, inputs can be divided based on the model's response to them, like classifying inputs based on whether the model answers them correctly (Yang et al., 2024; Højer et al., 2025).

**Prompting Template.** Next, the inputs are often formatted into a prompt that activates the concept's representations. Multiple papers use an A/B-choice format consisting of a question, two answer choices designated with "A:" and "B:", and the letter corresponding to the answer (Panickssery et al., 2024). Here, it is important to randomize the order, to not introduce the letter or position as a confounder. Another common template adds pre-prompts to a question that instruct the model towards or against the concept (Zou et al., 2023a).

**Differences Between Contrastive Sets.** Some methods construct contrastive pairs of inputs. These are derived from the same example but transformed into two contrasting inputs. For example, one can use the same question but attach a different pre-prompt (Zou et al., 2023a) or answer to it (Panickssery et al., 2024). Deng et al. (2025) employ a matched pair trial design where pairs of inputs differ only in the roles assigned to them and are guaranteed to lead to opposite behaviors. In other methods, the positive and negative inputs are unrelated to each other since they are not derived from the same example (Wu et al., 2024c; Xu et al., 2024b; Beaglehole et al., 2025). Generally, contrastive pairs are preferable since they better isolate the target concept from other factors that could differ between the sets of inputs. However, for some settings, contrastive pairs can be harder to obtain.

### 4.1.2 Collecting Activations

The provided inputs are fed into the model to perform a forward pass for which the activations $\text{Act}(\mathcal{M}, X^c, l, m, p) \to A^c_{l,m,p}$ are recorded at a specific layer $l$, model components $m$, and token positions $p$. It is possible to find the best options for these hyperparameters by testing the effectiveness of the resulting concept operator (Arditi et al., 2024).

**Layer.** It is a common practice to collect the activations from all layers and subsequently compute a concept operator for each layer. Alternatively, one can reduce computational cost by focusing on layers that are more likely to contain the concept's representation (Ghandeharioun et al., 2024).

**Model Component.** There are multiple distinct places within a transformer block at which activations can be collected. The most common ones are the residual stream after the MLP or the attention layer, the outputs of individual attention heads, and the outputs of an MLP or attention layer. Analyzing the MLP or attention heads gives insight into the computations done at that layer, while focusing on the residual stream is akin to analyzing the intermediate memory of a program.

**Token Position.** Some methods read the activations at a specific token position that has special significance towards the concept, commonly, the last token (Scalena et al., 2024) or the end-of-sequence token (Rozanova et al., 2023), the token containing an answer (Panickssery et al., 2024), or the token mentioning the concept. Other methods collect activations for all tokens and then take their average to serve as a representation of the sequence (Li et al., 2024e).

### 4.1.3 Calculating the Concept Operator

The activation collection step results in sets of activations that have different relationships to the concept. These can be used to calculate the concept operator. Assuming that we are given a positive and a negative set of activations $A^+_{l,m,p}, A^-_{l,m,p}$ with $n$ samples each, we need a function $f(A^+_{l,m,p}, A^-_{l,m,p}) \to o^c_{l,m}$ that calculates the concept operator.

**Difference in Means.** Taking the difference between the classes of activations can identify the direction that captures their differences. Difference-in-Means (DiM) is a popular method that simply calculates the average for both classes and subtracts the negative from the positive one (Panickssery et al., 2024; Jorgensen et al., 2024; Ball et al., 2024). This is equivalent to taking the average of the difference between pairs of examples $o^c = \frac{1}{n}\sum_{a^+,a^- \in A^+,A^-} a^+ - a^-$ . Earlier papers took the difference between a single pair of inputs (Turner et al., 2024; Liu et al., 2023), however, it has been found that taking the difference across larger datasets leads to higher-quality operators (Chu et al., 2024; Jorgensen et al., 2024). Arora et al. (2024) employ an unsupervised method that discovers 2 clusters of activations through $k$-means and subtracts their means. Im & Li (2025) show that DiM is optimal at identifying a vector that matches negative examples to positive ones. However, these vectors are not suited to steering positive examples where they lead to large performance reductions.

**Probes.** Probes are classifiers that predict whether activations belong to the positive or negative class $f_\theta(a) \to \{c^+, c^-\}$ (Alain & Bengio, 2018). The weights $\theta$ of a probe should capture the concept's representations. Many papers employ logistics regression classifiers as probes and use the learned weight vector as the concept operator (Chen et al., 2024a; Guo et al., 2024; Wang et al., 2024a). Contrast Consistent Search identifies a linear probe that retains logical consistency $P(c|x_i) = 1 - P(\text{not } c|x_i)$ (Burns et al.,

2023). Others train multiple orthogonal probes and combine them through a weighted sum (Chen et al., 2024b). Multiple papers use MLP-based probes, which are more expressive (Hoscilowicz et al., 2024; Singh et al., 2024). Wang et al. (2024c) use k-means clustering on the difference between positive and negative pairs to then train one linear probe per cluster, thus discovering multiple aspects of the concept. Xu et al. (2024b) find a vector that can be added to activations to prevent a linear probe from detecting the concept. However, Im & Li (2025) critique that probes only capture the direction and not the magnitude necessary to match positive and negative examples. Beaglehole et al. (2025) train a probe that learns non-linear features from the activations, before deriving a matrix that captures the directions in activation space to which the probe is most sensitive. The top eigenvectors of these matrices are then used as concept operators. Lastly, Nguyen et al. (2025) train concept vectors to match the negative to positive activations distribution using a Maximum Mean Discrepancy loss that captures higher order differences in distributions like variance.

**Dimensionality Reduction.** Dimensionality Reduction techniques can help to identify the most important information about the concept. It is common to apply Principal Component Analysis to the difference between the two sets of activations $PCA(A^+ - A^-) = \lambda_1, \lambda_2, ...$ before selecting the first principle component as the concept operator $o^c = \lambda_1$ (Zou et al., 2023a; Adila et al., 2024b). However, the direction with the highest variance does not necessarily capture the shift between positive and negative examples (Im & Li, 2025). Other papers use Singular Value Decomposition to identify the most important components of each group of inputs (Adila et al., 2024b), their difference (Tlaie, 2024; Ma et al., 2025), or their covariance matrix (Qiu et al., 2024). Xiao et al. (2024) criticize that dimensionality reduction techniques lose crucial information and instead train a Gaussian Mixture Model to model the distribution of positive and negative activations.

**Additional Techniques.** In addition to these techniques, multiple papers attempt to identify neurons or attention heads which are more relevant to the concept and then only calculate the operator from those. The idea is to find a minimal steering intervention that causes fewer side effects by assuming that the concept's representation is only encoded in the selected neurons. Li et al. (2023a) and Ma et al. (2025) only intervene on the attention heads where a probe achieves high accuracy, and Todd et al. (2024) focuses on attention heads that have a causal effect on a specified task. Other papers find important neurons by calculating attribution scores per neuron (Wu et al., 2024b), by selecting ones that exhibit low variance on the difference between sets (Li et al., 2024c) or by identifying causally influential nodes through Activation Patching (Xiao et al., 2024). Another technique first projects activations into a different space before deriving an operator. For example, Zhang et al. (2024b) train an autoencoder that projects activations into a space where they are easily separable.

## 4.2 Output Optimization

Representation Identification methods that perform Output Optimization provide information about the concept by scoring outputs. This is done by first providing inputs and collecting their activations and outputs. Secondly, a loss function is constructed by evaluating the outputs with regard to the concept. Thirdly, an optimization algorithm is used to iteratively optimize the activations to achieve a high concept score. In our running example this can be done by defining a way to score whether model generations are truthful and then optimizing a concept operator that can be applied to make truthful outputs more likely.

### 4.2.1 Collecting Activations

Output Optimization methods provide some inputs $X$ and collect the resulting activations that will be optimized $\text{Act}(\mathcal{M}, X^c, l, m, p) \to A^c_{l,m,p}$ at a specific layer $l$, model components $m$, and token positions $p$.

**Source of Inputs.** Inputs $X$ can be taken from existing datasets (Wu et al., 2024c), or be generated manually or synthetically (as in Section 4.1.1). While they do not need a specific relationship to the concept, they are usually meant to produce outputs that are relevant to the concept. The inputs can be a query (Yin et al., 2024; Cao et al., 2024a; Zeng et al., 2024) or a sentence that the model should transform (Subramani et al., 2022; Konen et al., 2024).

Table 3: Steps and differences between different **Output Optimization** methods for **Representation Identification**.

| Steps In Output Optimization | How do methods differ? |
|---|---|
| 1. Collecting Activations | **Source** of inputs (existing datasets, manually written, or synthetically generated)
**Layer** (all layers, or concept-related layers only)
**Model component** (residual stream, output of MLP or attention heads)
**Token position** (End of sequence, all tokens, or concept-related tokens) |
| 2. Output Scoring | **Output Format** (token, sentence, or long-form generation)
**Scoring** (dataset or scoring function)
**Loss Function** (Cross-Entropy, KL-divergence, regularization) |
| 3. Optimizing Activations | **Optimization Algorithm** (SGD, iterative sampling)
**Deriving Concept Operator** (combined training, individual operators, averaging) |

**Layers, Model Components, and Token Positions.** Next, the activations of the model that will be optimized are chosen. Activations are usually collected at every layer from the attention heads, residual stream, or MLP or attention layer outputs. However, some papers focus on attention heads for which a linear probe achieves high accuracy (Yin et al., 2024; Zeng et al., 2024). Furthermore, we need to decide at which token positions the optimization will be applied. While some papers only focus on the activations during generation (Konen et al., 2024), others use the activations on input tokens (Cai et al., 2024), a combination of input and output activations (Wu et al., 2024c), or activations of tokens with specific syntactic relevance to the concept, e.g. for the token naming a person (Hernandez et al., 2024).

### 4.2.2 Output Scoring

The model generates outputs $M(X) \to Y$ that are scored according to the concept $\text{Score}_c(Y)$, which is used as a loss function $\mathcal{L}_{\text{Score}}(Y)$.

**Output Format.** Based on the input, we can let the model generate a single token (Hernandez et al., 2024), a sentence (Subramani et al., 2022), or a long-form generation (Cao et al., 2024a; Wu et al., 2024a;c).

**Scoring.** These outputs are now evaluated on how much they align with the concept. This provides information about the concept to the optimization process. The outputs can be evaluated by comparing them to ground-truth answers from a dataset or through a scoring function. A score can be derived from the log-probability the LLM assigns to the ground-truth token (Hernandez et al., 2024; Wu et al., 2024c) or sentence (Subramani et al., 2022; Yin et al., 2024) or from the difference in log-probabilities between a desired and undesired answer (Cao et al., 2024a; Zeng et al., 2024). A scoring function can judge how much an output aligns with the concept. Cai et al. (2024) ask an LLM whether the output is aligned with the concept and use its token probabilities on "yes" and "no" as a score.

**Loss Function.** The score is then transformed into a loss function. The Cross-Entropy loss is popular (Subramani et al., 2022; Konen et al., 2024; Wu et al., 2024a;c). Yin et al. (2024) add a regularization term to induce sparsity and Hernandez et al. (2024) additionally minimize the KL-divergence compared to the unsteered token distribution to minimize unwanted side-effects. Other papers adapt previously developed loss functions for preference optimization like DPO (Cao et al., 2024a) or ORPO (Zeng et al., 2024). Lastly, Wu et al. (2025) suggest ReFT-r1, which optimizes a joint objective for the probing accuracy and steering effectiveness of a concept.

### 4.2.3 Optimizing Activations

A concept operator is optimized to modify activations so that they lead to outputs that score highly on the loss function $o^c = \text{argmin}_{o^c} \mathcal{L}_{\text{Score}}(\mathcal{M}(F(A, o^c)))$.

**Optimization Algorithm.** After we have attained a loss function with regards to the concept, we can employ an optimization algorithm to adapt the activations to minimize that loss. The most common method for this is using Gradient Descent with the AdamW optimizer (Subramani et al., 2022; Hernandez et al., 2024; Yin et al., 2024; Cao et al., 2024a; Wu et al., 2024a;c). Cai et al. (2024) propose an iterative algorithm that samples outputs before backpropagating to a steering vector that makes the highest scoring output more likely.

**Derive Concept Operator.** The concept operator is optimized for the loss function using the optimization algorithm, while the activations and weights are kept frozen. This concept operator should combine information about the concept from different outputs so that it generalizes to unseen examples. For a set of inputs and outputs, we can train one shared concept operator using SGD over batches of inputs and evaluated outputs (Yin et al., 2024; Wu et al., 2024c). Subramani et al. (2022) instead trains one concept operator for each input. However, this concept operator fails to generalize to different inputs. (Konen et al., 2024) overcome this by first deriving one concept operator per input before averaging them into a generalized concept operator.

## 4.3 Unsupervised Feature Learning

Representation Identification methods that perform Unsupervised Feature Learning (UFL) first learn sparse features in the model's activations and then identify which concepts they belong to. This is done by first collecting a set of activations, applying a learning algorithm that discovers features in the activation space, and then identifying which human-understandable concept a feature corresponds to. In contrast to other RI approaches, UFL methods do not set out to find one specific concept representation but identify a large set of concepts from which concepts of interest can be chosen. In our running example, we would first discover a library of representations for many concepts and then identify which one corresponds to truthfulness.

**UFL methods.** The dominant paradigm in UFL are Sparse Autoencoders (SAEs) (Templeton et al., 2024; Huben et al., 2024). They are motivated by the observation that a model represents more features than it has directions, thus forcing the model to represent features in superposition (Elhage et al., 2022). SAEs aim to disentangle these representations by learning a dictionary of monosemantic directions that represent sparse concepts. This dictionary receives the LLM's activations and is trained to accurately reconstruct them while only using a small set of its features at a time.

Deep Causal Transcoding (DCT) (Mack & Turner, 2024a;b) attempts to find a minimal perturbation in the activations of a layer that causes a large, predictable change in a later layer. As such, DCT might have applicability for eliciting latent behaviors like backdoors or unknown capabilities.

Table 4: Steps and difference between **Unsupervised Feature Learning** methods for **Representation Identification**.

| Steps in Unsupervised Feature Learning | How do methods differ? |
|---|---|
| 1. Collecting Activations | **Inputs** (large-general, small-specific datasets) 
 **Location** (residual stream, attention/MLP layer outputs) |
| 2. Discovering Features | **Goal** (disentangle features, or find meaningful perturbations) 
 **Loss Function** (reconstruction and sparsity, consistent large change) 
 **Learned Objects** (feature dictionary, diverse steering vectors) |
| 3. Identifying Concepts | **Identification** (highly activating, probing accuracy, mutual information) 
 **Labels** (LLM-as-a-judge, existing dataset) |

### 4.3.1 Collecting Activations

UFL methods provide inputs $X$ and collect activations $\text{Act}(\mathcal{M}, X^c, l, m, p) \rightarrow A^c_{l,m,p}$ at a specific layer $l$, model components $m$, and token positions $p$.

**Inputs.** SAEs employ large datasets to collect many activations. The dataset does not have to be specifically related to the concept of interest. However, SAEs require much larger datasets than other RI methods. This inefficiency partially stems from the fact that SAEs learn many features, while other methods focus on a single concept. DCT uses a much smaller set of inputs to collect activations. The inputs can be specific to a domain or have wide coverage and do not require a specific relationship to the concepts of interest.

**Location.** The activations for SAEs are usually collected from the residual stream but have also been taken from the outputs of the attention layers (Kissane et al., 2024) and MLP layers (Bricken et al., 2023). DCT collects activations from the residual stream. DCT uses a much smaller set of inputs to collect activations from the residual stream.

### 4.3.2 Discovering Features

An unsupervised learning algorithm is applied to discover features $o$ in the activations $A$.

**Goal.** SAEs are trained to take the model's activations, project them into a higher dimensional and sparse space, and then project them back. The goal is to disentangle the representations of concepts that are superimposed on each other. Since the dimensionality of the SAE is much larger than that of the LLM, it can represent each concept as a monosemantic direction while retaining the same information as the LLM.

The key idea of DCT is that perturbations along semantically meaningful directions at one layer will cause large, consistent, and predictable changes at later layers, whereas the effect of meaningless perturbations is inconsistent and quickly diminishes over the layers.

**Loss.** The SAE is trained to have a low reconstruction loss and to use sparse features. The reconstructed activations should be very similar to the original activations, and any feature in the SAE should only activate on a few inputs.

In DCT, a set of vectors is optimized over a set of inputs so that adding them to the activations at an early layer causes a large change in later layers. The vectors are optimized to have effects that a shallow MLP model can predict with high accuracy. Furthermore, orthogonality between vectors is enforced to find a diverse set of features.

**Learned Objects.** SAEs learn an MLP that projects from the LLM's activations into the SAE space and one MLP that projects back. The directions in the SAEs space can be seen as features in this learned dictionary. Ideally, they should denote representations of concepts within the LLM and can be used as steering vectors. DCT produces vectors that can be added to the model's activations to achieve specific downstream effects. They can be used directly as concept operators.

### 4.3.3 Identifying Concepts

Lastly, one needs to identify which concepts $c$ the learned features $o$ correspond to, thus deriving the concept operator $o^c$. This requires a method for identifying the features and a source of information about the concept.

**Identification.** One can inspect the inputs and outputs on which the feature activates most strongly. However, Durmus et al. (2024) find that the context in which a feature fires does not always predict its effect. Alternatively, one can increase the value of the feature and observe the effect on the model's outputs. Other methods identify feature directions that have a high mutual information with the target concept (Zhao et al., 2024), optimize a collection of features to lead to desired outputs (Kharlapenko et al., 2024), or select the features with the highest probing accuracy for a concept (Wu et al., 2025)

**Labels.** Early works viewed highly activating inputs and outputs manually to determine the corresponding concept. Current work leverages automated interpretability by using LLMs to make these judgments. Other

methods require a pre-existing dataset labeled for the concept. For example, Wu et al. (2025) build a dataset of positive and negative examples for a concept to determine the probing accuracy of features.

### 4.3.4 Applying SAEs

Once one has used SAEs to identify a feature that represents the targeted concept, the direction of that feature can be used as a concept operator. However, using that concept operator for steering requires one to translate every activation into the SAE space and back. Therefore, there have been attempts to use SAE features to improve steering vectors in the LLMs activations that were identified by other methods. `Chalnev et al. (2024)` derive steering vectors that activate desired SAE features. Conmy & Nanda (2024) decompose a steering vector into SAE features and remove features that seem unrelated to the desired concepts. However, this is challenging, as the steering vectors fall outside of the SAEs training distribution and contain negative feature directions which are not accommodated in the SAE (Mayne et al., 2024). Kharlapenko et al. (2024) address this by optimizing the SAE reconstruction of the steering vector to achieve good downstream performance while incentivizing sparsity, leading to cleaner reconstructions.

Notably, SAEs have recently been extended to discover circuits of features implemented over adjacent layers (Dunefsky et al., 2024) or multiple layers (Lindsey et al., 2024). Furthermore, there has been continued progress for better SAE architectures (Rajamanoharan et al., 2024a;b). This progress will hopefully lead to better methods for identifying and steering LLMs.

Many human-understandable concepts, such as famous people or scam emails, have been identified through SAEs (Templeton et al., 2024). However, the method does not guarantee to find a specific concept of interest. An engineer might be interested in the representation for honesty but fail to find one in the SAE. Furthermore, while many papers motivate SAEs with the ability to steer representations, few of them actually evaluate whether the discovered features allow for effective Representation Control. To improve the applicability of SAEs for feature steering, it should be a standard to evaluate the effects of features on models' outputs.

---

**Takeaway: Representation Identification**

To identify the representation of a concept, methods need to provide a stimulus related to the concept, collect the model's activations, and calculate a concept operator from them. There are three types of methods:

1. **Input Reading**: Construct sets of inputs related to the concept, collect their activations, and calculate the concept operator from the sets of activations.
2. **Output Optimization**: Collects activations and outputs, scores the outputs according to the concept, and optimizes a concept operator for that score.
3. **Unsupervised Feature Learning**: Collect activations, discover features in the activations with an unsupervised learning algorithm, and identify which concepts the features correspond to.

---

## 5 Representations Operationalization

RepE assumes that the model represents the concept of interest in its activation space. However, it is unclear how exactly the model's representations are structured. RepE methods for identification and control bake in assumptions about the geometry of representations used by the model. These representations are operationalized in RepE methods with a concept operator that has a specific shape and intended relationship to the models' representation of the concept.

### 5.1 Assumed Geometry

Assumptions about the geometry of internal representations are included in the choice of RI method and steering function. For example, if the weights of a linear probe are used as a concept operator, it will not be able to pick up on non-linear aspects of a concept's representation. Similarly, if an activation steering function only increases along a linear direction, it would fail if the concept was represented along a non-euclidean space.

Table 5: Differences between approaches for operationalizing representations (the output of the **Representation Identification** step).

| Aspects of Representation Operationalization | How do methods differ? |
| --- | --- |
| Assumed Geometry | **Features as** Linear Direction, Fuzzy clusters or Non-linear manifolds |
| Choice of Concept Operator | **Shape** (Vector, Matrix, Vector + Matrix) 
 **Relationship** with concept representation (Exact match, Capture covariance, Capture principled direction and variance) |

There is an ongoing discussion in the interpretability community about the geometry that transformer models use to represent internal features. The Linear Representation Hypothesis poses that concepts are represented as linear directions in the activation space, and their intensity is denoted by the magnitude of the activations in that direction (Park et al., 2024b). Although this hypothesis has some merits and, if true, would make RepE easier, it has also been criticized (see Section 12 for discussion). Thus, RepE methods with alternative feature geometries have been proposed. They assume representations to be fuzzy, clusters (Postmus & Abreu, 2024), or non-linear manifolds (Qiu et al., 2024). Other possibilities for feature geometries include circles (Engels et al., 2025), onions (Csordás et al., 2024), non-linear paths, or subsets of a vector space. See Section 13.2.2 for consequences of these assumptions and Section 14.2 for possibilities for extension.

## 5.2 Concept Operators

The concept operator is the object that denotes the model's representation of the concept. As the output of RI and the input for RC, it is the linking element between the two parts of the RepE pipeline. Concept operators can take different forms like vectors, matrices, or combinations thereof. Furthermore, they differ in the relationship they are intended to have to the model's representation.

**Vectors.** The most common type of concept operator is a vector $o^c \in \mathcal{R}^d$, where $d$ is the dimensionality of the activation space. Often, this is used by methods that assume that representations are linear directions in the model's activation space, which a vector can describe. Thus, the concept operator vector should be as similar as possible to that direction. Other methods follow Li et al. (2023a) and employ one vector per attention head (Guo et al., 2024; Wang et al., 2024c; Li et al., 2024d). Furthermore, some methods denote a concept representation through a set of vectors that represent different aspects of the concept (Chen et al., 2024b) or serve different functions (Wu et al., 2024a). Cai et al. (2024) use a set of vectors that were optimized to steer activations towards specific outputs to train a LoRA-adapter that produces similarly steered activations.

**Matrix.** A range of matrices have been used as concept operators $o^c \in \mathcal{R}^{d \times d}$. Postmus & Abreu (2024) operate on the assumption that features are fuzzy clusters in the activations. They use a positive semi-definite matrix called a conceptor that should capture the principle directions and variances. This allows to more precisely steer complex representations since conceptors also capture the underlying correlations between activations. Xiao et al. (2024) two Gaussian Models that respectively model the distribution of positive and negative activations. This retains information contained in the distribution of activations while allowing activations to be mapped from one distribution to the other. Qiu et al. (2024) employ 2 matrices that capture covariance with positive and negative activations, respectively. These matrices can be used to project new activations to maximise their covariance with positive demonstrations and minimize covariance with negative demonstrations. Lastly, Rozanova et al. (2023) learn a Nullspace Projection Matrix $o^c$ which combines nullspaces that are orthogonal to the feature direction $o^c a = (I - (o^c_0, ..., o^c_n))a \to a^c$. This captures information in the activations that a probe could use to predict the concept of interest. Thus, the matrix can be used to project out information about the concept.

**Matrix + Vector.** Some methods employ both a matrix and a vector to denote the concept representation, most commonly to perform an affine transformation where the activations are multiplied with the matrix and

then added to the vector (Dong et al., 2024b). Singh et al. (2024) and Avitan et al. (2024) find that matching the mean and covariance of negative and positive activations allows one to steer the model's representations. Thus, they derive a matrix that captures the covariance of the concept's representation and a vector that captures its mean. This allows to steer the representation with an affine function. Wu et al. (2024c) extend this by learning an additional matrix that serves to project the activations in and out of a space where the affine transformation is more effective. Hernandez et al. (2024) employ a matrix to amplify directions of an attribute that are relevant to a concept entity and a bias term that better fits the transformed attribute into the activations. This assumes representations of entities and attributes to be represented linearly. Alternatively, Pham & Nguyen (2024) assume that representations are directions of activations from the origin and that the magnitude along that direction reflects the intensity of the representation. They use the weights vector of a probe that separates the positive and negative regions in the activation space and an MLP that can predict how to adjust the angle of an activation vector.

---

**Takeaways: Representation Operationalization**

1. **Representaiton Geometry**: RepE methods make implicit assumptions about the geometry of representations. Many assume that concepts are represented as linear directions, but alternative perspectives exist.
2. **Concept Operator**: Most commonly, the concept operator is a vector denoting a linear direction. Others use matrices to capture principled directions, covariances or all linearly available information about a concept.

---

## 6 Representation Control

The goal of Representation Control (RC) is to steer the representations of the LLM in order to control its outputs. For this, it employs a **steering function** that makes use of the concept operator to modify the weights or activations of the model. Additionally, RC methods decide when and where to apply the steering functions. The choice of concept operator $o^c$ and steering function $F$ determine the effect of RC and bake in assumptions about the geometry of model representations.

Throughout the section, we will describe the proposed steering functions along with the concept operators they use. We describe what effect the steering function has on the activation space and how this is supposed to control the representations of the model.

### 6.1 Modifying Activations

Activations can be controlled at inference time with an activation steering function $F_a(a_l, o^c_l) \rightarrow a^c_l$ (see Definition 4). For a given input, we read the activations at an intermediate layer $a_l$ during the forward pass, perform some operation $F$ on it using the operator $o^c_l$, and then use the modified activations $a^c_l$ as input for the next layer. By steering the activations, we can control the model's output with regard to the concept. In our running example, we would take activations and steer them to be more similar to truthful activations and less similar to dishonest ones, thus causing later computations and the outputs to be more honest.

#### 6.1.1 Activation Steering Functions

**Linear Addition.** This simply adds a vector $o^c \in \mathbb{R}^d$ to the original activations $a_l + \lambda o^c_l \rightarrow a^c_l$, where the factor $\lambda$ determines the steering strength. The model's representation is assumed to be a linear direction, and the concept operator is a vector that should denote that direction. Linear Addition simply increases the magnitude of activation along the direction, which should increase the intensity of the concept in the model's representations.

Table 6: Differences between **Modifying Activations** methods for **Representation Control**.

| Aspects of Modifying Activations | How do methods differ? |
|---|---|
| Steering Functions | **Operation** (Linear Addition, Vector rejection, Affine transformation, ...) 
 **Concept Operator** (vector, matrix, vector+matrix) 
 **Assumed Geometry** (linear directions, mean&covariance matching) 
 Intended Effect on **Activation Space** (scale direction, remove information, rotate direction) 
 Intended Effect on **Representations** (increase/decrease intensity, remove) |
| Location & Time | **Layer & Model component** (single or multiple layer(s) & component(s)) 
 **Token position** (every token, one token before generation, specific token) 
 **Input dependent** (independent, switch on/off, adjust intervention strength) |

Linear Addition is the most popular steering function. Improving on this simple function, some papers renormalize the activations after the addition to a normal magnitude, thus avoiding reductions in quality (von Rütte et al., 2024; Adila et al., 2024b; Liu et al., 2024b; Leong et al., 2023; Chen et al., 2024b). Other methods employ a set of vectors that are applied to individual attention heads (Li et al., 2024d; 2023a). Following Differential Privacy, Zeng et al. (2024) adds noise to the vector to protect privacy.

Multiple papers attempt to add multiple steering vectors at once to steer multiple concepts or multiple aspects of one concept. This can be done by simply adding all vectors at once (Wang et al., 2024c; Scalena et al., 2024) or combining them in a weighted sum (Chen et al., 2024b; Zeng et al., 2024). However, this leads to interference between concepts and increases negative impact on the LLM's quality (see Section 13.1). van der Weij et al. (2024) avoid this by adding different vectors at different layers. Adila et al. (2024b) decompose the different vectors into their orthonormal basis and use those for steering.

**Vector Rejection.** The vector $o^c$ can be rejected $a_l - \frac{a_l \cdot o_l^c}{o_l^c \cdot o_l^c} o_l^c \rightarrow a_l^c$, by first projecting $a_l$ onto $o_l^c$ before subtracting. This removes the component of $a_l$ that is parallel to $o_l^c$, leaving us with the part of $a_l$ that is perpendicular to $o_l^c$. Assuming linear feature directions, this removes the influence of the concept $c$ in the representations (Arditi et al., 2024; Deng et al., 2025). Equivalently, vector scaling $a_l + \frac{a_l \cdot o_l^c}{o_l^c \cdot o_l^c} o_l^c \rightarrow a_l^c$ amplifies the components of the activations that are aligned with the concept operator (Chu et al., 2024).

**Nullspace Projection.** This employs a Nullspace projection matrix $o^c$ that captures information that a probe would use to classify the concept in the activations. Multiplying activations with this matrix removes the information of that concept $o_l^c a_l \rightarrow a_l^c$ (Rozanova et al., 2023), thus removing the influence of that concept on the model's outputs.

**Soft Projection.** This multiplies the activations with a conceptor matrix (discussed in Section 5.2) that encodes the principal directions and variances of a concept set of activation vectors. This scales the activation vector along the directions relevant to the concept while not completely overwriting the non-concept directions. Thus, it allows for a more nuanced control over representations (Postmus & Abreu, 2024).

**Rotation.** Pham & Nguyen (2024) argue that rotating activations into the appropriate direction from the origin is superior to shifting the direction directly since rotations achieve the desired direction while retaining the magnitude of activations. For computational efficiency, they approximate such a rotation through a reflection followed by an adjustment of the angle. Undesired activations are reflected along a hyperplane into the desired region of the activation space. Next, a learned MLP determines how to adjust the angle of the reflected activations to attain the desired activation.

**Affine Transformations.** Affine Transformations for activation steering employ a matrix $o_l^{c,M}$ and a vector $o_l^{c,v}$ to control the activation $a_l$ through an affine transformation $o_l^{c,M} a_l + o_l^{c,v} \rightarrow a_l^c$. Singh et al. (2024) apply affine steering to match the mean and covariance of two representations. They prove this as an optimal

steering function since it guards against linear probes for that concept, thus removing any linearly available information about the concept. Similarly, Xiao et al. (2024) recenter and then rescale activations to fit into the distribution of positive activations. Affine Steering has been used to manipulate facts the model associates with an entity (Hernandez et al., 2024), project a steering vector into the activation space of another model (Dong et al., 2024b), or to create counterfactuals with regard to sensitive attributes (Avitan et al., 2024).

**Projecting Before Editing.** Another approach is to project activations into a space that allows for better editing. Qiu et al. (2024) first project into a non-linear space, before they project activations into directions that co-vary maximally with positive demonstrations and minimally with negative ones. Arora et al. (2024) project different activations onto the feature vector to then subtract them without influencing other features.

In section 4.3 we discussed that SAEs can learn vectors that correspond to interpretable concepts. However, these vectors are in the representation space of the SAE. If one wants to use them to steer the normal model, it is possible to project the model's activations into the SAE space, apply the steering on the feature of interest, and then project them back. However, this comes with larger decreases in capabilities due to the reconstruction error and at higher computational cost.

### 6.1.2 Location & Time

**Layers and Components.** Representation Control methods can modify the activations at different layers or components. While some methods only steer the activations at a single layer and component (Panickssery et al., 2024; Zhang et al., 2024a; Cao et al., 2024a; Ball et al., 2024), others modify activations on many or all layers and components (Konen et al., 2024; Chen et al., 2024a; Liu et al., 2024a; Adila et al., 2024a; Qiu et al., 2024). The optimal location for steering can be found by measuring steering effectiveness (Zhang et al., 2024a), probing accuracy (Zhang et al., 2024b), inspecting the projection of each layer's steering vector to the vocabulary (Cao et al., 2024b), or specific loss functions (Wang & Shu, 2023; Adila et al., 2024a).

**Tokens.** Furthermore, one could steer at different token positions. Some papers decide to steer at every token position during generation (Bortoletto et al., 2024; Adila et al., 2024a; Wang & Shu, 2023). While this can be more effective (Subramani et al., 2022), it also increases the impact on capabilities. Others only steer at the last token of the query, hoping that the steering effects propagate throughout the generation (Wang et al., 2024c; Adila et al., 2024b; Li et al., 2024c). Some papers chose to steer at or before a token that gives a prediction or answer (Paulo et al., 2024; Lucchetti & Guha, 2024).

**Dynamic Activation Steering.** For most activation steering functions, the intervention does not depend on the input. This is suboptimal since some inputs might require different steering directions or strengths. Dynamically adapting activation steering to the present context is an emerging theme within RepE since it can increase steering effectiveness and reduce unwanted side effects.

Multiple papers decide whether steering is necessary for an input based on the cosine similarity between the activations and the steering vector (Adila et al., 2024a; Wang et al., 2024b), between the activations and the average of steered activations (Ma et al., 2025) or depending on the output of a probe given the activations (Stickland et al., 2024; Li et al., 2024d; Pham & Nguyen, 2024; Nguyen et al., 2025). Similarly, the intervention decision can be made separately for each layer or component (Zhang & Viteri, 2024; Simhi et al., 2024). Instead of a binary decision for intervention, other papers modulate the strength of the intervention. This can be based on cosine similarity (Cao et al., 2024b), KL-divergence between steered and unsteered output (Scalena et al., 2024), the L2-norm of positive and negative activations (Leong et al., 2023), or by trying different strengths and evaluating the outputs (Fatahi Bayat et al., 2024). When multiple operators are available, the weight assigned to each operator can be derived dynamically (Zeng et al., 2024; Wang et al., 2024c). Lastly, the operation itself can be dynamic, like using an MLP to predict the angle by which the activations are rotated (Pham & Nguyen, 2024).

Table 7: Differences between **Modifying Weights** methods for **Representation Control**.

| Approaches to Modifying Weights | How do methods differ? |
| --- | --- |
| Training weights | **Training Target** (activations and/or outputs) 
 **Trained weights** (low-rank adapter, all weights) |
| Editing weights | **Operation** (weight orthogonalization, addition) |

## 6.2 Modifying Weights

The representations of a model can be steered by modifying the weights with a weight steering function $F^w(w_l, o_l^c) \rightarrow w_l^c$. In our running example we would modify the weights of the model so that it produces activation patterns and outputs that are aligned with honesty.

### 6.2.1 Training Weights

The weights can be modified through fine-tuning. This training is either aimed at producing desired activations or optimizes for specific outputs. Furthermore, fine-tuning methods differ in which weights or adapters they train.

**Training Towards Desired Activations.** Weights $w_l$ can be trained to produce desired activations $a_l^c$ by a weight steering function. A dataset of desired activations can be gathered by applying an activation steering function $F_a(a_l, o^c) \rightarrow a_l^c$. Zou et al. (2023a) optimize a low-rank adapter (LoRA) to generate activations with low L2-distances to steered activations. Similarly, Cai et al. (2024) train a LoRA adapter on a dataset of optimally steered activations so that the resulting activations are similar to the optimally steered activations. Alternatively, Zhang et al. (2024c) design an Adversarial Training setup where a LoRA adapter is trained to generate activations that a discriminator classifies as aligned with the concept.

**Training Representations without a Concept Operator.** Some weight steering functions do not rely on a concept operator $o^c$. Instead, they train weights to produce activations similar to random noise. Zou et al. (2024); Li et al. (2024b); Rosati et al. (2024) fine-tune an LLM so that activations on harmful inputs or information that should be unlearned are close to noise. This should remove harmful representations from the model.

**Training Towards Desired Outputs.** Multiple methods propose to fine-tune a small number of added parameters towards a dataset of desired outputs. Wu et al. (2024c) propose ReFT, which trains an adapter to bring the activations into a subspace where a linear projection and addition are applied. Yin et al. (2024) fine-tune only the bias terms of selected attention heads. Wu et al. (2024a) train an adapter consisting of a scaling and a bias vector. While these methods are inspired by RepE and aim to control the model's representations, they are also end-to-end parameter-efficient fine-tuning methods like LoRA adapters (Hu et al., 2022). This is shown by Jiang et al. (2024b), who draw a theoretical connection between LoRA and ReFT and unify them into a meta-algorithm for model efficient fine-tuning. This highlights that weight steering functions that optimize for specific outputs can be seen as being on a spectrum between RepE and fine-tuning.

**Training Towards Desired Activations and Outputs.** Other methods optimize both for desired activations and desired outputs. Ackerman (2024) optimize weights to produce activations with high cosine similarity to the concept operator vector while also retaining a low cross-entropy loss in the outputs. Yu et al. (2024) repeatedly ablate the vector that represents refusal while training the model to refuse harmful answers. This should force the model to learn more robust representations of that concept. Stickland et al. (2024) mitigate performance loss by training a LoRA adapter that minimizes KL-divergence in outputs when a steering vector is applied to the outputs of the original unsteered model.

### 6.2.2 Editing Weights

It is also possible to edit the weights directly without training. Arditi et al. (2024) apply weight orthogonalization $w_l - o_l^c o_l^{c^T} w_l \to w_l^c$ to the MLP weight matrix that writes to the residual stream. This modifies the component weights to never write in the direction $o_l^c$. If $o_l^c$ captures the direction with which the model represents a concept, this operation removes the concept from the model's representations. Wang et al. (2024a) simply add the weights of a probe $o^c$ to a few selected row vectors within the gated projection matrix of the MLP weights. This shifts activations in the hidden state of the MLP away from undesirable regions and towards desirable regions.

> **Takeaway: Representation Control**
> Representations can be controlled by steering the activations or weights.
> 1. **Activations** can be steered during inference by addition, rejection, or projection, thus increasing, removing, or adapting the concept representation. Activation steering can be applied at different tokens and dynamically adjusted for new inputs.
> 2. **Weights** can be steered by training them to generate desired activations and outputs or by editing components of weight matrices.

## 7 Practical Representation Engineering Pipelines

In this section, we describe some prototypical RepE pipelines and classify how they fit into our taxonomy. We compare them according to selected criteria. In addition, we gather empirical evidence to answer which design choices in RepE work best.

### 7.1 Prototypical RepE Pipelines

In Table 8, we classify RepE pipelines according to which strategy they employ for Representation Identification, the shape of their operator, and whether they control activations or weights. We score Representation Identification methods on whether they require a dataset labeled according to the concept and whether they require expensive optimization. Additionally, we score whether the method is restricted to linear representations or can capture non-linear ones. Furthermore, we score whether controlling the model requires retraining of its weights and whether the operation to control the model is dependent on the given inputs. Lastly, we determine the types of concepts the method was initially designed to engineer. However, this does not mean that they are not able to engineer other concepts. We select these RepE pipelines to show a diversity of approaches while preferring highly-cited papers that were published at top conferences.

**Contrastive Activation Addition (CAA).** CAA (Panickssery et al., 2024) provides positive and negative examples of a behavior, collects their activations, and takes the mean difference between the 2 sets of activations to calculate a vector. This vector is used as a concept operator that is added to the activations during inference, thus steering the model with regard to high-level behavioral characteristics.

**Low-Rank Representation Adaptation (LoRRA).** LoRRA (Zou et al., 2023a) identifies the operator by adding positive or negative pre-prompts to inputs, collecting their activations, and taking the first principle component of the difference between the two sets of activations. Instead of directly adding the vector to the activations, they train a LoRA adapter to output activations that are similar to adding the vector to the original activations. This can steer a range of high-level concepts, including honesty, emotions, and ethical values.

**Inference-Time Intervention (ITI).** ITI (Li et al., 2023a) inputs questions and answers that do or do not correspond to the concept. Then, a linear probe is trained on the activations of each attention head to predict whether the answer was or was not related to the concept. They select the attention heads with the highest probing accuracy and use the weight vector of the probe as a concept operator. During inference,

Table 8: Prototypical methods for RepE classified according to our taxonomy and the criteria they meet.

| Name | Representation Identification | | | Representations Operationalization | | Representation Control | | | Originally proposed for |
|---|---|---|---|---|---|---|---|---|---|
| | Type | Works with unlabeled dataset? | Optimization Free? | Concept Operator | Allows non-linear representations? | Type | Training-free? | Context-adjusted steering? | |
| CAA (Panickssery et al., 2024) | Input Reading | ✗ | ✓ | Vector | ✗ | Activations | ✓ | ✗ | High-level behaviors |
| LoRRA (Zou et al., 2023a) | Input Reading | ✓ | ✓ | Vector | ✗ | Weights | ✗ | ✗ | High-level behaviors |
| ITI (Li et al., 2023a) | Input Reading | ✗ | ✗ | Vectors | ✗ | Activations | ✓ | ✗ | Truthfulness |
| SAE (Templeton et al., 2024) | Unsupervised Feature Learning | ✓ | ✗ | Vector | ✗ | Activations | ✓ | ✗ | Abstract features |
| LoReFT (Wu et al., 2024c) | Output Optimization | ✓ | ✗ | 2 Matrices + vector | ✗ | Activations/ Weights | ✗ | ✓ | Improving task performance |
| BiPO (Cao et al., 2024a) | Output Optimization | ✗ | ✗ | Vector | ✗ | Activations | ✓ | ✗ | Aligning to preferences |
| SEA (Qiu et al., 2024) | Input Reading | ✗ | ✓ | 2 matrices | ✓ | Activations | ✓ | ✓ | High-level behaviors |
| MiMiC Singh et al. (2024) | Input Reading | ✗ | ✓ | Matrix + vector | ✗ | Activations | ✓ | ✗ | Removing bias & Toxicity |

these vectors are added to the activations of their respective attention heads. This method was first proposed to make models more truthful.

**Sparse Auto-Encoder (SAE).** SAEs (Templeton et al., 2024) are trained in an unsupervised fashion on a large dataset of activations to learn internal features. Activations are projected into a wider feature space, which is optimized so that features are sparse and can be used to reconstruct the original activations. We can identify the concept a feature corresponds to by investigating inputs or outputs for which the feature is highly activated. These features correspond to vectors in the embedding space of the SAE and can be used to stimulate the model's activations. SAEs find a wide range of concepts at different levels of abstraction.

**Low-rank Linear Subspace Representation Finetuning (LoReFT).** LoReFT (Wu et al., 2024c) learns an adapter that intervenes with the activations during a forward pass. The adapter first projects the activations into a linear subspace, edits the activations with an affine transformation, and projects them back. The outputs of the adapter are added to the original activations. For this, a low-rank matrix, projection matrix and bias vector are trained to minimize the cross-entropy loss on a dataset that exemplifies the concept. This method was proposed to improve performance on specific downstream tasks.

**Bi-directional Preference Optimization (BiPO).** BiPO (Cao et al., 2024b) optimize a vector so that adding the vector to the activations makes the desired outputs more and the undesired outputs less likely. Subtracting the vector should have the opposite effect. After optimizing with Gradient Descent, we get a vector that represents the preferences in the provided dataset. During inference, the model's representations are controlled by adding the vector to the activations. This was proposed to efficiently align LLMs with human preferences.

**Spectral Editing of Activations (SEA).** SEA (Qiu et al., 2024) collects activations for a neutral question and for corresponding positive and negative responses. By applying Singular Value Decomposition on the covariance matrices between negative and neutral, as well as positive and neutral activations, they identify a positive and negative editing matrix. During inference, they can select directions in the activations that co-vary with the editing matrices and prune negative directions while retaining positive ones. They extend this to non-linear directions by first transforming the activations into a non-linear space where the editing

can be performed before transforming the edited activations back. While SEA is not designed for particular types of concepts, it was initially used to steer truthfulness and bias.

**Minimally Modified Counterfactual (MiMiC).** MiMiC (Singh et al., 2024) divides activations into a positive and negative set based on an MLP probe trained to detect the concept. They then derive an affine steering function that matches the mean and the covariance of the two sets of activations while causing minimal distortion. On new activations, this function performs a projection and translation, steering it towards the target concept while preserving key statistical properties. It was initially proposed to control representations of toxicity and bias in sensitive concepts.

## 7.2 Which Methods Work Better?

In this section, we summarize previously reported results that compare different methods. While this section gathers existing empirical evidence for comparison between RepE methods and implementations, we often cannot make conclusive statements about the superiority of one method. This is because of a lack of unbiased comparisons (see Section 14.5 for discussion).

### 7.2.1 What Works for Representation Identification?

**Is it more effective to use Input Reading or Output Optimization? - Depends on the concept.**
Multiple papers claim that their method based on output optimization improves on input reading methods (Yin et al., 2024; Cai et al., 2024; Cao et al., 2024a; Ackerman, 2024). However, we lack evidence for the contrary, which is likely because papers that propose input reading techniques did not compare their methods to output optimization methods. Two unbiased comparisons find that an input reading method is better at steering the sentiment of text (Konen et al., 2024) and that output optimization methods control the concrete topics more effectively (Wu et al., 2025).

**Are steering features from Sparse Auto-Encoders better than other RI methods? - Possibly a combination.**
In Zhao et al. (2024) and Makelo et al. (2025), an SAE-based steering is more effective than input-reading-based RI methods. In contrast,Chalnev et al. (2024) find that an input-reading-based RI method leads to more effective steering than SAE features. However, results in Chalnev et al. (2024) and Kharlapenko et al. (2024) suggest that SAEs can be used to make input-reading-based concept operators more effective. Finally, evaluations on the AxBench (Wu et al., 2025) find that SAEs perform worse at concept detection and steering than LoReFT and class means but better than methods using probing or dimensionality reduction.

**Which prompting format works best? - Unclear.**
Braun et al. (2025) compare control effectiveness for seven prompt settings varying in whether an answer token is appended, whether an instruction is prepended, and whether few-shot demonstrations are included. While the setting that only appends an answer token has a slightly larger average control effectiveness, they conclude that no prompt type clearly outperforms the other. Wang & Shu (2023) find that choice prompts, where the model has to select answer A or B, outperform free-form answers. This is likely because the choice prompt concentrates the relevant context into a single token and thus more effectively triggers the concept representation. The lack of principled evaluations of optimal prompting formats for eliciting representations makes this an important area of further study.

**At which token positions should activations be read out? - Unclear.**
Some evidence suggests that averaging the activations of multiple tokens is better than just using the last token's activations (Hoscilowicz et al., 2024). Furthermore, focusing on the last token given by the user can be more effective than taking activations from an appended post-prompt like "I think the {attribute} of this user is" (Chen et al., 2024a). Braun et al. (2025) do not find a clear difference between reading activations at the last token of the question or at the token containing the answer.

**Which function is best for calculating the difference between sets of activations? - Difference-in-Means.**
In Input Reading, functions like Difference-in-Means (DiM), linear probes, Principle Component Analysis, or Contrast Consistent Search (Burns et al., 2023) are used to identify the concept representation from

contrasting sets of activations. This has been experimentally studied in 7 papers (Li et al., 2023a; Marks & Tegmark, 2023; von Rütte et al., 2024; Xu et al., 2024a; Arora et al., 2024; Wu et al., 2025; Im & Li, 2025). In 6 out of 7 papers Difference-in-Means leads to the strongest performance. The next best seems to be taking the weight vector of a linear probe. PCA and CCS tend to perform worse. A fair comparison between DiM, PCA, and classifier-based methods finds that DiM is best at positively and negatively steering multiple-choice questions, equals the performance of probes at steering long-form generations, but leads to the largest performance deterioration on positive examples (Im & Li, 2025). Lastly, Distributed Alignment Search (DAS) (Geiger et al., 2024b) was only tested once in which it performed the strongest, highlighting that DAS remains underexplored. Besides Input Reading, it is advantageous to apply mean-centering to the steered activations as shown by Jorgensen et al. (2024).

### 7.2.2 What Works for Representation Operationalization?

**Are matrices or vectors better concept operators? - Matrix more effective but more costly.**
Multiple works that propose using matrices instead of vectors as concept operators find that they improve steering effectiveness (Postmus & Abreu, 2024; Rajendran et al., 2024; Pham & Nguyen, 2024). Results from Zou et al. (2023a) are more ambivalent, finding that a matrix slightly outperforms a vector-based concept operator, but is clearly outperformed by a vector that is derived from additional forward passes for every new input. While matrices can be more effective, they can also come at an increased cost (Postmus & Abreu, 2024) or reduced output quality (Rajendran et al., 2024).

**Are linear or non-linear concept operators better? - Non-linear.**
While most methods perform linear operations on the representations, the direct comparison between linear and non-linear methods favors the non-linear ones. Non-linear Spectral Editing of Activations (SEA) outperforms linear SEA (Qiu et al., 2024), and non-linear ITI is more effective than ITI, although this comes at a higher cost in capabilities (Hoscilowicz et al., 2024).

### 7.2.3 What works for Representation Control?

**Is it more effective to modify weights or activations? - Unclear.**
This question is still open. Wang et al. (2024a) find that their method for controlling model weights is more effective than intervening on activations while leading to a larger reduction in output quality. However, a broader evaluation of that question is necessary.

**Which steering function should be used for modifying activations? - Unclear.**
Zou et al. (2023a) find that a piece-wise operation is more effective than linear addition. Chu et al. (2024) find that projection outperforms a product, which again is more effective than linear addition. Luo et al. (2024) find that linear addition is more effective than orthogonal projections at a higher cost in capabilities. Linear addition remains the most popular activation steering function despite some other steering functions outperforming it. This warrants further investigation comparing different operations. Krasheninnikov & Krueger (2024) study different activation steering functions in a toy classification setup. They find that more expressive steering functions have a higher performance ceiling and can control more complex tasks, but simpler steering functions are effective with lower amounts of data.

**At which token positions can activations be steered most effectively? - Apply steering at every inference step.**
Overall, it is more effective to continuously apply RepE at every inference step than to only apply it at the first token (Subramani et al., 2022). This is because the steering effect diminishes throughout the generation (Scalena et al., 2024). However, only intervening on the first token can still be very effective (Subramani et al., 2022) and has a lower impact on output quality (Scalena et al., 2024).

**Which component of the transformer should be steered? - Unclear.**
RepE often modifies activations in the residual stream, the attention or the MLP block, but it is unclear which component is most effective to steer. Existing evaluations only find that steering at the LM head or embedding layer is ineffective (Subramani et al., 2022) and that steering both the attention and MLP blocks is more effective than only intervening on one of them (Zhang et al., 2024b).

Table 9: Source of labels and metrics for evaluating LLMs on different generation

| Form of generation | Free generation | Likelihood on specified answers |
|---|---|---|
| **Source of Labels** | LLM-judge, text classifier, word occurrences | correct answers, ground-truth text |
| **Metrics** | Judge Score, # of concept-related words | Accuracy, Logprob-based metrics |

**Should only one or many layers be controlled? - Many layers.**
It is generally assumed that intervening on all layers at once leads to more effective steering at a higher cost in the general capabilities of the model. However, to the best of our knowledge, there are no experiments proving this. Notably, many concepts can be most effectively steered in middle layers (Panickssery et al., 2024; Im & Li, 2025).

## 8 Evaluation of RepE Methods

Good practices and resources for evaluations are a key drivers of progress for any area of Machine Learning. However, for RepE evaluations are not standardized and metrics are still developing. This section outlines current and best practices for evaluating RepE methods and describes existing and potential future benchmarks.

### 8.1 Common Evaluation Methodologies

The general setup for evaluating RepE methods involves measuring the steering effect, a measure of how much the behavior of model with regard to the concept of interest was changed. For this we use the model before and after intervention to generate outputs on a dataset or task that is often related to the concept. For the generated outputs we use a metric that determines in how much the outputs are aligned with the concept. Additionally it's common to evaluate in how much the general model quality deteriorates.

#### 8.1.1 Evaluation Task and Dataset

Depending on the concept of interest, different tasks and datasets are used. It is common to use established datasets and benchmarks that cover the steered concept or targeted downstream application. Commonly used datasets for evaluating changes in a concept are TruthfulQA (Lin et al., 2022) for increasing truthfulness, AdvBench (Zou et al., 2023b) for preventing harmful outputs or BOLD (Dhamala et al., 2021) for reducing societal bias (see Section 10 for more common datasets per concept). Other papers devise their own specific tests and dataset, for example to measure causal efficacy of an identified concept operator (Arora et al., 2024), to evaluate steering effect a wide range of concepts (Wu et al., 2025) or to study RepE in a toy setting (Krasheninnikov & Krueger, 2024).

A key difference between evaluations is whether the model is asked to generate answers to multiple-choice questions or make longer generations. While evaluations on multiple-choice questions are convenient to run, they are less faithful to downstream settings in which a model will be employed (Pres et al., 2024).

#### 8.1.2 Measuring Steerability

To evaluate the steering effect, it is necessary to measure in how much a model's outputs are aligned with the concept. This requires a way to label outputs depending on their alignment with the concept and to construct a metric out of this. A key difference in evaluations of the steering effect is if the model gets to freely generate any output that then gets evaluated or whether we are evaluating the likelihood of the model generating specific pre-defined answers.

**Source of Labels** Evaluation methodologies differ on how they determine the amount of concept alignment of an output. Sources of labels can be LLM judges, ground-truth labels or text of a datasets, and word occurrences in generated text.

LLM judges are commonly used to evaluate freely generated text. Many papers instruct a frontier LLM such as GPT-4 to give a numerical judgment about how much a generated text aligns with the concept of interest (Wu et al., 2025). Other papers use smaller models that are specialized for judging a specific property and have been validated for this task. For example Wang & Shu (2023) use small, specialised LLM for toxicity classification (Caselli et al., 2021) and a lexicon and rule-based tool for sentiment scoring (Hutto, 2020).

Many papers use exiting labels in a dataset. In multiple-choice benchmarks such as MMLU, the correct labels can be used as ground truth to judge the models performance. Furthermore, on a texts related to the concept, the ability to correctly predict the correct next token can be taken as a source of information about the concept alignment of a model (van der Weij et al., 2024). Other papers judge outputs based on how often specific concept-related words occur. Human judgments are rarely used, since they are expensive. Only Wang et al. (2024c) use a small set of human evaluators to confirm that their automated metrics align with human judgment.

**Metrics for the Steering Effect** Depending whether pre-specified or freely generated outputs are used, different metrics for the steering effect are available.

Evaluation methodologies that use a judge for scoring usually simply take the score assigned by the judge as their metric. For example LLM judges are often instructed to provide a score (e.g. from 0-4), while specialized judging-models might directly output a concept score.

Some concepts can also be measured by detecting or counting concept-related words and phrases in freely generated outputs. A common technique for measuring the Attack Success Rate of jailbreaks is to measure the look the presence of specific words and phrases such as "I am sorry as an LLM..." (Arditi et al., 2024). Turner et al. (2024) count the number of wedding-related words and Jorgensen et al. (2024) count the number of words related to specific words to measure whether they were able to shift the topic and genre of generated text.

On multiple-choice questions many papers measure the average accuracy. On longer generations, van der Weij et al. (2024) measure the difference in the top-1 accuracy at predicting the correct next token on datasets that are or aren't related to the concept. However, this ignores shifts in the probability the model assigns to the correct answer. Thus, Panickssery et al. (2024) evaluate steering effect as the average likelihood the model assigns to the correct response. Tan et al. (2024) instead use the logit-difference propensity, which is calculated as the difference in logits between the positive and negative answer. They argue that excluding the normalization provided by the last softmax makes the metric more linear with respect to the model's activations. Alternatively, Turner et al. (2024) measure the probability of the correct answer being in the top-k tokens. Arora et al. (2024) devise a log-odds metric that measures the causal efficacy of an intervention on the model's representation. Krasheninnikov & Krueger (2024) count the fraction of examples where the model can change the desired attribute without changing any other attributes. Finally, Pres et al. (2024) take the difference in log-likelihoods for positive and negative continuations, normalize them and then take the mean difference between the base and steered model.

### 8.1.3 Measuring Changes in Model Quality

Interventions on the models representations should have minimal impact on the general quality and capabilities of the model. RepE papers use a variety of measures to evaluate their impact on model quality.

For this, many papers report scores on popular capabilities benchmarks like MMLU (Hendrycks et al., 2021) or MT-Bench (Zheng et al., 2023a) as a measure of models quality. Other papers evaluate the models capability to correctly predict the next token by running it on a general dataset like the Pile (Gao et al., 2020) and evaluating Perplexity(Scalena et al., 2024), top-1 accuracy (van der Weij et al., 2024) or the probability to have the correct token in the top-k tokens (Turner et al., 2024). Furthermore, it is possible to evaluate specific aspects of model performance that could be deteriorated by RepE. Brumley et al. (2024) measure diversity of generations bi- and tri-gram entropies. Wu et al. (2025) evaluate the fluency and instruction following of the model via an LLM-judge.

Lastly, some papers decide to combine steering effectiveness and model quality into one performance metric for RepE. von Rütte et al. (2024) propose perplexity-normalized effect size, which divides the increase in

probability of the concept being present by the decrease in model fluency. Wu et al. (2025) report the harmonic mean of the score for concept-alignment, instruction following and fluency as a combined metric.

### 8.1.4 Other Experimental Details

**Hyperparameter Ablations:** RepE methods are commonly evaluated on a range of hyperparameters and settings. For many RepE methods the steering strength is a hyperparameter that guides how strong the intervention is. Altering the steering strength can outline the tradeoff between steering effect and model quality. Furthermore, this indicates the quality of the concept operator is, since increasing the steering factor should lead to a monotonic or even linear increase in concept-alignment (Pres et al., 2024; Tan et al., 2024). Other common hyperparameters to be altered are the sets of layers on which the intervention is conducted, the token position to intervene on, the number of samples used in Representation Identification and the family and size of models.

**Baselines:** To measure the steering effect and reduction in model quality the steered model must be compared to the original model without intervention. However, we also need to compare to other control models. It's common to compare to earlier RepE methods. Furthermore, many papers make comparisons to prompting or fine-tuning methods.

## 8.2 Best Practices for Evaluating RepE Methods

**Free generation vs likelihood of specified answers.** Measuring the steering through measures that use pre-specified answers has the advantage that one can measure shifts in the log-likelihood of the model. Pres et al. (2024) argue this is important because disregarding the confidences of the model, loses information about how variable behavioral expressions are. On the other hand, when none of the pre-specified answers has a high likelihood, such metrics would not take into account the actual generations a model would produce thus miss important aspects of its behavior.

**Using specialized judges.** Many studies rely on LLM-as-a-judge setups, where a highly capable and general model is instructed to rate generated text. However, using LLMs-as-a-judge is known to be biased, sometimes inaccurate and is usually not scientifically validated as a metric (Fu et al., 2024; Zheng et al., 2023b). When possible it might be preferrable to use specialised models that have been developed and validated to score a specific concept, such as small LLMs trained for detecting specific concepts (Inan et al., 2023; Caselli et al., 2021; Dementieva et al., 2023).

**Varying Steering Strength.** In many RepE methods it is possible to modulate the strength of the intervention. By showing the steering effect and model quality at different steering strengths readers can get a sense of the tradeoff between these properties. Furthermore, it is useful to plot the relationship between steering strength and steering effect. If the steering effect monotonically, linearly or strongly increases with higher steering strength this showcases that the RepE can effectively modulate the desired concept. Tan et al. (2024) even propose a metric based on this relationship. For the same reason it is informative to show the effect if a negative steering strength is applied.

**Realistic and Difficult Test Settings.** While it is convenient to use multiple-choice format to evaluate the steering effect, Pres et al. (2024) argue that it is important to the effect on open-ended generations since this setting is more similar to downstream settings where RepE will be applied. Additionally, it is preferable to evaluate RepE on difficult settings that are similar to real world deployment. For example, when using RepE to defend against jailbreaks it is important to use state-of-the-art attacks and evaluate multi-turn jailbreaks.

**Reporting Changes in Models' Quality.** Only a minority ( 35%) of papers report the change in models' capabilities after applying Representation Engineering. Since we know that RepE can reduce these capabilities, it is important to measure the LLMs' quality to compare the merits of each method. Furthermore, it is desirable to administer diverse capability tests, like measuring the ability to generate fluent text while also the evaluating the accuracy at answering multiple-choice questions.

**Measuring Generalization.** RepE methods can struggle to generalize to different settings (see Section 13.1). Thus, it is not sufficient to evaluate the steering effect on another subset of the dataset used for identifying the concept operator. Instead studies should systematically vary aspects of the settings between Representation Identification and Control.

**Comparing to Strong Baselines.** When proposing a new method, its effectiveness should be compared to the current state-of-the-art RepE method. However, often papers only compare to early methods that have since been surpassed. Similarly, when comparing to other families of methods, studies should not compare to a naive prompting or fine-tuning approach, but to the state-of-the-art method for their respective problem.

**Evaluating Diverse Concepts.** The effectiveness of RepE methods should be measured on a diverse range of concepts. Since steerability can differ between concepts it is difficult to assess the quality of a method on only one concept. Furthermore, it is desirable to evaluate on different types of concepts (see Section 9.1) to determine whether a method has strengths or weaknesses at steering some type of concept.

**Testing on Different Model Sizes.** Appendix A.1 finds that most RepE experiments are conducted on models between 3-10 billion parameters. To prove the general effectiveness of a RepE method, it is necessary to evaluate it on models of different sizes. Specifically, larger models need to be tested to indicate that the method scales to frontier models.

**Survivorship Bias.** It is rare for papers that propose new methods for Representation Engineering to demonstrate concepts or situations in which control is not effective. However, papers that set out to neutrally evaluate the method often find weaknesses or limitations not obvious from the original paper. This practice can give false perceptions and stifle progress in the field. Thus, we encourage authors to report on null results.

**Reporting Variance.** Tan et al. (2024) rightly point out that many papers only report on average steerability. However, it is essential to showcase the distribution of results so readers see whether a method provides reliable steerability.

## 8.3 RepE benchmarks

**Existing Benchmarks.** There are two benchmarks that attempt to evaluate and compare RepE methods (Wu et al., 2025; Im & Li, 2025).

AxBench (Wu et al., 2025) is a benchmark that evaluates the ability of RepE methods to identify concept operators that accurately detect the presence of a concept and steer the LLMs' behaviour. For this, they built a synthetic dataset of positive and negative examples for 500 concepts. Using this dataset, 9 RepE methods are used to derive concept operators and compared to prompting and fine-tuning. Steering ability is determined by generating responses to instructions while performing Representation Control and then using an LLM to score the generation on presence of the concept, instruction following and fluency of the generation. For concept detection Difference-in-Means, Logistic Regression and ReFT-r1 perform best. However, prompting outperforms all RepE methods at steering LLM behaviour to a concept. Notably, all tested concepts are of a similar type, testing whether a concrete, narrow topic is contained in a text. This benchmark could be extended by consider different types of concepts.

Im & Li (2025) evaluate four methods for Representation Identification on their ability to steer outputs positively and negatively towards a concept on multiple-choice and open-ended generation tasks. They find that Difference-in-Means outperforms other methods to identify a steering vector. This benchmark could be extended by comparing different methods for Representation Operationalization or Control and evaluating them on a more diverse range of concepts.

**Desireable Benchmark Properties.** There is currently no benchmark that provides a way to comprehensively evaluate the effectiveness of a RepE pipeline. Thus we are currently unsure which pipeline is most effective in which situations. A benchmark can also serve as a guide for the future development of RepE methods by providing clarity about the effectiveness of methods and being a target to aim for. Such unified evaluations can also surface previously undiscovered limitations of methods (Brumley et al., 2024).

Such a benchmark could also provide a testing ground for rigorous comparison. One challenge in developing a RepE benchmark lies in providing fair comparisons between RepE methods that have different requirements, assumptions, and goals. They might have different requirements on the available data, require different amounts of compute, or were developed to control different kinds of concepts.

An ideal benchmark for RepE should cover a range of concepts of different types since different RepE methods might steer different concepts more effectively (Brumley et al., 2024). Hereby, the focus should lie on use cases where RepE is foreseen to be practically useful, such as improving truthfulness, changing goals, adapting situational awareness, or removing societal biases. Furthermore, evaluations should cover multiple models, test the influence of dataset size, and show the impact of different steering strengths. Additionally, it should specifically evaluate the weaknesses of current RepE methods (see Section 13.1), thus putting a spotlight on improvements on these challenges. Of course, this benchmark should make sure to follow best practices outlined above (Section 8.2).

## 9 What Concepts Can Be Controlled with Representation Engineering?

### 9.1 Types of Concepts

To illuminate how RepE can be applied, we cluster the concepts that have been controlled using RepE into six categories of concepts. Table 10 showcases that most RepE papers steer high-level behavioral concepts and that RepE can be successfully applied to all types of concepts.

Table 10: The number of papers that attempt to, successfully or not, steer a type of concept.

| Type of Concept | Success | Failure |
|---|---|---|
| High-level behavioral characteristics | 56 | 8 |
| Tasks | 11 | 4 |
| Language and style | 10 | 0 |
| Knowledge and beliefs | 9 | 1 |
| Content | 8 | 0 |
| Values, goals, and ethical principles | 7 | 1 |
| Linguistic or grammatical features | 6 | 0 |

**High-level behavioral characteristics** like harmfulness or honesty are the most popular category of concepts controlled via RepE. These concepts are fairly abstract, may be context-dependent and cannot easily be specified. They are in reference to the behavior exemplified by the model.

**Tasks** a model should carry out, including tasks like reasoning, classification and associations can be induced by RepE. Such concepts specify an input-output function, for example, by finding a function vector that triggers the execution of a task (Todd et al., 2024).

**Language and style** of a text, such as sentiment, style, genre, or language can be controlled with RepE. Such concepts are high-level properties of the generated text.

**Knowledge and beliefs** encoded in a model about the world can be controlled with RepE. Such concepts reference what the model assumes to be true while generating text. This includes changes to the model's factual knowledge or its situational beliefs (see Section 10.3).

**Content** of the model's outputs are amendable to RepE, like steering it to focus on a specific topic and contain or not contain some contents. These contents are often not exact text, but higher-level concepts like "talking about weddings" or "social security numbers". For example, the Concept500 dataset (Wu et al., 2025) was designed for evaluating RepE and contains positive and negative examples for 500 concrete topical concepts. Such concepts are properties of the generated text.

**Values, goals, and ethical principles** that are encoded in an LLM have been adjusted with RepE. This refers to overarching objectives and principles according to which the LLM is making its decisions. It has

been applied to implementing human preferences into the LLM's behavior (Cao et al., 2024a) or adapting which ethical theories are used for decision-making Tlaie (2024).

**Linguistic or grammatical features** like verb conjugations or the gender of nouns are controlled in multiple papers. Such concepts are specific to language, and their presence can easily be verified in the generated text. While controlling them is not of great interest itself, they are used as a concept for interpretability (Hao & Linzen, 2023) or to evaluate RepE methods (Arora et al., 2024).

## 9.2 Commonly Controlled Concepts

This section showcases concepts for which RepE has been applied most often. For this, we selected all steered concepts in the surveyed papers. Table 11 shows the concepts which have been controlled in $\geq 5$ papers and how many of the experiments were successful or failed in steering that concept. Here, success is defined as providing a significant improvement in the measure of the concept compared to the uncontrolled model.

Table 11: Commonly steered concepts along with the amount of experiments where the concept was or was not successfully controlled.

| Target Concept | Success | Failure |
| --- | --- | --- |
| Truthfulness | 17 | 1 |
| Harmfulness | 17 | 0 |
| Toxicity | 16 | 0 |
| Fairness | 11 | 0 |
| Refusal | 10 | 1 |
| Sentiment | 7 | 0 |
| Privacy | 5 | 0 |

The most common concepts are Truthfulness, Harmfulness, and Toxicity. This is because they are of practical importance in the context of LLM chatbots and because RepE seems well suited to control them, but also because early seminal papers experimented on these concepts and later work followed. Furthermore, we see a very high success rate for all concepts. However, this statistic should be viewed critically since experiments that fail to control a concept are less likely to be published. Nevertheless, it does provide evidence that these concepts can be controlled effectively.

> **Takeaway: What concepts have been controlled with RepE?**
> There are different types of concepts that can be controlled with RepE, among which high-level behavioral concepts like Truthfulness, Harmfulness, and Fairness are the most popular.

## 10 Applications of Representation Engineering

After describing some popular concepts that can be effectively steered with RepE, this section describes concrete problems for which RepE has been applied.

### 10.1 AI Safety

AI Safety aims to prevent harm caused by or with AI models. In the context of LLMs this often refers to preventing outputs that are harmful to the user, the use of LLMs for harmful applications, or untruthful and hallucinated statements. We also include the prevention of private data leakage and the refusal behavior of LLMs as safety concerns. Furthermore, the field studies threats of future advanced AI models such as deception or self-improvement in current LLMs.

**Reducing Harmfulness.** Harmfulness is a broad category describing outputs that can cause some harm to users, other individuals, or society. It includes more granular categories, such as toxic outputs, wrong

information, or private data leakage. To control harmfulness in general, one needs to identify and steer harmful representations (Cai et al., 2024; Zou et al., 2023a; Scalena et al., 2024; Chu et al., 2024; Beaglehole et al., 2025; Deng et al., 2025). This approach can be combined with other safety techniques such as safe-decoding in a defense-in-depth approach (Banerjee et al., 2024). RepE can control harmfulness in multi-modal LLMs (Wang et al., 2024b), and harmful representations identified in one language can be transferred to another (Xu et al., 2024a). Furthermore, the effect of controlling the model's exhibited personality traits on the harmfulness of its outputs has been explored using RepE (Ghandeharioun et al., 2024; Zhang et al., 2024a).

Toxicity is a more specific concept referring to offensive, aggressive, or inappropriate outputs. RepE has been used to decrease toxicity (Wang et al., 2024a; Qian et al., 2024; Singh et al., 2024; Luo et al., 2024; Jorgensen et al., 2024; Pham & Nguyen, 2024; Li et al., 2024d; Turner et al., 2024; Chu et al., 2024; Nguyen et al., 2025). Concept operators for toxicity have been transferred from smaller to larger models (Dong et al., 2024b). However, by increasing the intensity of the toxicity representation, RepE can also be used to make models more toxic and induce attacks on LLMs (Wang & Shu, 2023).

A range of datasets are used to operationalize harm with AdvBench (Zou et al., 2023b), HarmfulQA (Bhardwaj & Poria, 2023) and HarmEval (Banerjee et al., 2024) being among the most popular. RepE methods tend to prevent harmful outputs more effectively than prompting or fine-tuning-based safeguards (Wang et al., 2024b; Cao et al., 2024b; Li et al., 2024c; Yu et al., 2024).

**Preventing and Triggering Jailbreaking, Refusal, and Backdoors.** Training LLMs to refuse to answer harmful requests is a key mechanism for ensuring that they do not produce harmful and toxic outputs. To stress-test these safety mechanisms, jailbreaks have been developed that can circumvent this refusal behavior (Yi et al., 2024). By directly manipulating the internal mechanism that triggers refusal, RepE can serve as a very potent technique to avoid overrefusal or ensure that refusals are effective and appropriate.

Wang & Shu (2023) were the first to show that adding a steering vector during inference undermines safety by reducing truthfulness, making outputs more biased and increasing toxicity and harmfulness. This was followed by a range of work that uses RepE to jailbreak models (Wang et al., 2024a; Ball et al., 2024; Xu et al., 2024b; Stickland et al., 2024; Li et al., 2024c; Tran et al., 2024; Zhang et al., 2024c).

Arditi et al. (2024) find that there is a single direction in the activations that mediates whether refusal behavior is triggered or not. Yu et al. (2024) ablate the refusal direction while performing safety training, forcing the model to learn more robust refusal behavior. Furthermore, RepE has been used to stop the LLM from refusing to answer benign queries (Cao et al., 2024b; Xiao et al., 2024).

Lastly, RepE has been used to identify backdoors through UFL (Mack & Turner, 2024a) and trigger backdoors by inducing the presence of the backdoor trigger (Price et al., 2024).

**Making LLMs Truthful.** A model is truthful if it outputs true answers. Getting models to give truthful and honest answers is crucial for their performance, but it also increases the trust we can place in them.

Li et al. (2023a) were the first to steer truthfulness using a RepE method. Marks & Tegmark (2023) find linear probes that can steer the model to treat false statements as true and vice-versa. Overall, RepE methods seem to be particularly suited for steering this concept, often outperforming fine-tuning (Ackerman, 2024; Chen et al., 2024b; Qian et al., 2024; Qiu et al., 2024; Li et al., 2023a; Liu et al., 2024a) although it is sometimes outperformed by prompting (Chen et al., 2024b; Li et al., 2023a; Wang et al., 2024c). The initial methods have been extended to encapsulate multiple aspects of truthfulness (Wang et al., 2024c; Chen et al., 2024b), steer non-linear representations (Hoscilowicz et al., 2024) or steer with different intensities before selecting the most truthful output with a probe (Fatahi Bayat et al., 2024). TruthfulQA is the most commonly used dataset (Lin et al., 2022).

Relatedly, hallucinations have been mitigated or induced with RepE (Wang et al., 2024c; Simhi et al., 2024; Zhang et al., 2024b;c; Panickssery et al., 2024; Beaglehole et al., 2025). Simhi et al. (2024) find that hallucinations can be more effectively identified from activation before the model answers and more effectively controlled in the activations of attention heads.

**Increasing Privacy.** LLMs are trained on large amounts of uncurated text from the internet that contains private information, which could be leaked by the LLM.

To mitigate these privacy concerns, RepE has been used to prevent private data leakage (Zhang et al., 2024a; Qian et al., 2024; Cai et al., 2024). Wu et al. (2024b) find that securing specific private data can increase the leakage of other private information, which they successfully suppress by controlling privacy-sensitive neurons. Zeng et al. (2024) remove private information from the inputs and apply steering vectors to restore performance. In the tested setups, these techniques can even improve over Differential Privacy methods (Zeng et al., 2024; Wu et al., 2024b).

**Avoiding Future Risks.** The AI Safety community has hypothesized future risks from advanced AI models, and researchers are now attempting to study key aspects of these risks in LLMs.

A deceptive system could lie to humans about its intentions. While it is possible to make models more truthful in general, Clymer et al. (2024) are not able to uncover and steer deception. However, sycophancy, which is the tendency of LLMs to adjust their responses to what the user wants to hear, has been successfully controlled (Stickland et al., 2024; Panickssery et al., 2024; Templeton et al., 2024), although others fail to reproduce this (Paulo et al., 2024; van der Weij et al., 2024). AI systems might develop misaligned motivations, but this could be mitigated by engineering the representations of their goals (Mini et al., 2023). RepE has controlled an LLMs' exhibited desire for survival (Panickssery et al., 2024), power (Zou et al., 2023a), wealth (van der Weij et al., 2024), and short-term rewards (Panickssery et al., 2024; van der Weij et al., 2024). Furthermore, Panickssery et al. (2024) control corrigibility and willingness to coordinate with other AIs against humans. Additionally, understanding the capabilities of a model is important for estimating the risk it poses. Mack & Turner (2024a) propose a method that could elicit undiscovered capabilities through unsupervised feature learning.

## 10.2 Ethics

Many ethical questions surround the development and application of LLMs. These include concerns about societal biases and the fairness of models, aligning the LLM to the values of its users, and encoding ethical reasoning into LLMs.

**Improving Fairness.** Since LLMs are trained on text from the internet, they are prone to encode and amplify existing societal biases. An LLM might reproduce stereotypes, not be representative in its outputs, or treat members from different social groups differently. By identifying how and where specific biases are represented, RepE can aid in uncovering, showcasing, and mitigating such biases.

RepE methods have been used to identify representations of bias-sensitive attributes such as gender, age, or race and control them to reduce the bias in outputs (Pham & Nguyen, 2024; Chu et al., 2024; Wang & Shu, 2023; Nguyen et al., 2025). RepE has been used to control the sentiment expressed towards social groups (Luo et al., 2024) or how often a group is associated with a specific disease (Zou et al., 2023a). Singh et al. (2024) identify representations for gender and dialect bias and removes the bias by steering all inputs to the same gender or dialect class. Furthermore, Durmus et al. (2024) use SAEs to identify features for gender, age, and political bias. They also find a "Neutrality" and "Multiple Perspectives" feature that reduce bias across many dimensions of bias.

Additionally, RepE can be used to steer the representation of protected attributes to create counterfactual outputs, which can surface biases and thus be used for auditing the model. Chen et al. (2024a) do this by controlling the model's belief about the user's age, gender, educational level, and socioeconomic status, thus uncovering how the model responds differently to different users. Avitan et al. (2024) intervene in representations of gender to create counterfactual generations.

Commonly used datasets for identifying the representations of biases include BOLD (Dhamala et al., 2021), BBQ (Parrish et al., 2022), or BiasInBios (De-Arteaga et al., 2019) datasets.

**Aligning with human preferences.** Pre-trained LLMs do not necessarily act according to the preferences and values of their users. To mitigate this, LLMs are commonly fine-tuned with methods like RLHF or DPO,

where humans give feedback to align the model with their preferences. Instead of fine-tuning the weights, we could also use RepE to align with human preferences.

Cao et al. (2024a) train a steering vector using a DPO-based loss that makes it more likely for the model to produce desired responses. It is also possible to use synthetically generated preference data to identify and steer relevant models' representations (Liu et al., 2023; Adila et al., 2024a).

**Changing Ethical Reasoning.** To aid in making LLMs act ethically, it is important they understand human ethical reasoning developed over millennia. RepE can help to improve general ethical reasoning (Pham & Nguyen, 2024). It can also steer the model towards specific moral theories like commonsense morality, utilitarianism, or deontological reasoning (Tlaie, 2024; Zou et al., 2023a; Xu et al., 2024a).

## 10.3 Knowledge Editing

LLMs encode a wealth of knowledge in their representations. However, some of that knowledge might be wrong, outdated, or dangerous. Thus, we would like to be able to remove and edit knowledge. Relatedly, LLMs can encode certain beliefs about the world or other systems that can be useful to interpret and edit.

**Editing Factual Associations.** Model editing methods like ROME (Meng et al., 2022) have successfully changed factual knowledge stored in LLMs. Similarly, Zou et al. (2023a) were able to make an LLM output that the Eiffel Tower is located in Rome. Hernandez et al. (2024) propose REMEDI, which can detect and edit the attributes associated with an entity through its activations. Furthermore, it is possible to use RepE to improve the ability of an LLM to reason with newly provided information (Yin et al., 2024). Zhao et al. (2024) do not edit factual associations but rather control whether an LLM uses its learned parametric knowledge or the contextual knowledge provided in the prompt to answer a question. Relatedly, it is possible to detect and steer how much the model reproduces memorized text from the training dataset (Zou et al., 2023a).

**Unlearning.** Unlearning is concerned with removing harmful or unwanted knowledge in a model. Li et al. (2024b) propose Representation Misdirection for Unlearning (RMU) that unlearns hazardous knowledge by pushing harmful representations towards a random vector while retaining harmless ones. However, Xu et al. (2024b) find that RMU and other unlearning methods are not robust against harmful steering vectors. Farrell et al. (2024) use SAE steering to unlearn hazardous knowledge but find that it is not yet a competitive unlearning method. Rozanova et al. (2023) are able to remove information about specific features. These mixed results indicate that RepE might become a useful tool for unlearning but currently still has weaknesses.

**Modulating Situational Awareness.** During inference, LLMs may hold beliefs about the situation they are currently in. These could be facts about the environment or other agents, which RepE can potentially identify and control. Chen et al. (2024a) identify and control the models' beliefs about the age, gender, educational level, and socioeconomic status of the user it is currently interacting with. Similarly, Ghandeharioun et al. (2024) control the model's inferred impression of the user's personas in order to elicit harmful responses. Price et al. (2024) change what the model believes the current year is. Lastly, RepE has been used to improve the model's capability to reason about the beliefs of other agents in Theory of Mind tasks (Bortoletto et al., 2024; Zhu et al., 2024).

## 10.4 Task Execution

Making LLMs carry out specific tasks is key to making them practically useful. This is usually approached through fine-tuning on task examples, by providing few-shot examples, or by giving precise instructions. Recently, RepE has also found applications to this by editing the representations to control which task is executed in what way.

Todd et al. (2024) identify a function vector (FV) that captures an ICL task by finding attention heads that have a causal effect on that task. In contrast, Liu et al. (2024b) find an in-context vector (ICV) from the activations on contrastive few-shot prompting inputs. Brumley et al. (2024) compare these two methods and find that ICVs are better able to steer behavioral tasks, while FVs control functional tasks more effectively. Furthermore, FVs generalize better and deteriorate model quality less. Jorgensen et al. (2024) improve the

ICV methodology by adding mean centering and Li et al. (2024e) apply inner and momentum optimization to the in-context vector. These works indicate that In-Context Learning partially works by shifting the representations to the correct task. Additionally, it is possible to identify representations that trigger the model to carry out Chain-of-Thought reasoning without being prompted to do so (Zhang & Viteri, 2024).

## 10.5 Controlled Text Generation

Controlled text generation involves generating text with specific attributes or constraints, such as sentiment, style, or topic. The challenge lies in balancing the generation of coherent, natural language while maintaining control over these characteristics. By shifting internal representations, RepE allows control over the desired attributes.

**Controlling Sentiment.** Sentiment is the emotional tone or opinion of a text. Researchers have used RepE to control how negative or positive the tone of generated text is (Turner et al., 2024; Konen et al., 2024) and what emotions are expressed in it (Cai et al., 2024; Konen et al., 2024; Zou et al., 2023a). Commonly used datasets are GoEmotions (Demszky et al., 2020) and the Yelp sentiment dataset.

**Controlling Personality.** It is also possible to shift the personality traits exemplified by the model. This can also indirectly influence other properties. Zhang et al. (2024a) steer personality traits of a model according to the MBTI-scale and study the effects on its safety properties. Weng et al. (2024) are able to steer the model towards OCEAN personality traits. By moderating the personalities, they are able to improve reasoning, enhance conversational capacity, and reduce sycophancy. Furthermore, it is possible to control the level of honesty and creativity displayed by the model with RepE (von Rütte et al., 2024).

**Controlling Language, Style, and Genre.** Guo et al. (2024) use input pairs from different languages to improve coherence in cross-lingual information retrieval and Scalena et al. (2024) find that safety steering vectors derived in English transfer to other languages. RepE can control the style of generated text, like writing Shakespearean or Chinese text (Konen et al., 2024; Beaglehole et al., 2025; Ma et al., 2025) and its genre, like fantasy, sci-fi, or sports (Jorgensen et al., 2024). It can also steer generation towards specific topics (Makelo et al., 2025; Templeton et al., 2024; Turner et al., 2024).

## 10.6 Performance

RepE has been applied to improve the performance of LLMs on general capabilities and specific tasks.

**Improving Reasoning.** Improving the reasoning abilities of LLMs is an important frontier in NLP research to which RepE has been applied (Cai et al., 2024; Wu et al., 2024c; Yin et al., 2024; Højer et al., 2025). RepE has been used to identify the representations underlying the Chain-of-Thought (CoT) mechanism and to control the model to produce CoT reasoning without being prompted to do so (Hu et al., 2024; Zhang & Viteri, 2024). However, an attempt to steer the faithfulness of CoT reasoning with regard to the actual decisions made by the LLM was unsuccessful (Tanneru et al., 2024). Additionally, RepE has been used to improve social reasoning abilities (Bortoletto et al., 2024; Zhu et al., 2024).

**Other Performance Improvements.** The capability of LLMs for Natural Language Understanding and Generation tasks has been improved through RepE (Wu et al., 2024c;a). van der Weij et al. (2024) use RepE for improving the quality of general code and of python-specific code and Lucchetti & Guha (2024) apply RepE to achieve type predictions that are more robust to irrelevant changes in the code. Furthermore, RepE has been used to make the model provide equivalent answers for semantically equivalent queries (Yang et al., 2024) and reduce its bias to the ordering of examples and answer options (Adila et al., 2024b). Guo et al. (2024) improve the quality of a multi-lingual information retrieval system by steering for coherence and accuracy. Lastly, Rahn et al. (2024) identify that uncertainty in the decision-making of an LLM agent is related to the entropy of its activations, which can be steered to achieve better exploration behavior.

## 10.7 Interpretability

Interpretability aims to help humans understand the internal processes of AI models. RepE has helped to identify how human-understandable concepts are represented and to study the impact of representations on outputs.

**Finding Linear Representations.** If a linear vector detects a concept with high accuracy and controls the concept effectively, this is evidence that this direction is how the model represents the concept. Thus, researchers have claimed to find linear representations of concepts such as Truthfulness (Marks & Tegmark, 2023), Refusal (Arditi et al., 2024), models' encoded beliefs about the current year (Price et al., 2024), subject number for conjugations (Hao & Linzen, 2023), and Harmfulness (Xu et al., 2024b). These results also provide some evidence for the Linear Representation Hypothesis (Park et al., 2024b). However, we should retain appropriate skepticism about evidence provided by RepE about the linearity of concept representations (see Section 13.2.1).

**Other Insights.** Aside from interpreting the shape of representations, RepE has been used to shed light on many phenomena in LLMs.

Wolf et al. (2024) identify a trade-off between helpfulness and safety. Qian et al. (2024) use RepE to investigate how safety-critical concepts emerge throughout pre-training. Rozanova et al. (2023) study the representations of safety concepts in different languages. They find them to be similar and see that English safety concepts can be used to steer other languages. Other papers investigate the inner mechanisms of CoT reasoning (Hu et al., 2024) and the impact of removing a concept (Rozanova et al., 2023).

However, RepE is not a perfect tool for interoperability, and results can easily be overclaimed. Wang & Veitch (2024) specifically criticize the idea that editing representations at a location which leads to the desired output change actually provides evidence that the concept representation is localized there. This is because they see that optimal edits at random points are also very effective.

## 10.8 Representation Engineering outside LLMs

### 10.8.1 Image Generation

A large number of previous works studied controlling image generation by operating on learned representations. Upchurch et al. (2017) showed that image editing can be done by linear interpolation in deep feature space between images with a certain attribute and images without that attribute, given the hypothesis that ConvNets linearize the manifold of natural images. Rich structures also appear in unsupervised representation learning of Generative Adversarial Networks (GANs). This can enable controlled generation by simple vector arithmetic (Radford et al., 2016) or non-linear traversal (Wang et al., 2021) operations on latent vectors to manipulate semantic concepts. Concept classifiers for attributes can be used to disentangle the representations of attributes, thus allowing us to constrain the optimization in a GAN to find directions that maximally affect the concept attribute (Wang et al., 2021). Shen et al. (2020) and Wu et al. (2021) showed that the latent space of StyleGAN2 can be disentangled so that each dimension captures at most one attribute. Others methods train GANs with contrastive learning to learn disentangled representations (Shoshan et al., 2021). Recently, such concept disentanglement in the activation space has been attempted in LLMs by training Sparse Auto-Encoders (SAEs) (Huben et al., 2024).

**Modulated Features.** Controlled image generation and editing is able to control attributes of a human, such as age, facial hair, eyewear, headwear, pose, facial expressions, and other global features, such as illumination and artistic styles (Shoshan et al., 2021; Bhattad et al., 2024). Collins et al. (2020) perform spatially localized edits based on a concept image. Jahanian et al. (2020) showed that activations can be steered to control camera movements.

**Analogy to LLMs.** Some techniques in controlled image generation resemble those in RepE for LLMs. Contastive learning to promote disentanglement of features (Shoshan et al., 2021) uses contrastive examples where an attribute is or is not present, which is similar to Input Reading with contrastive examples. Other methods use a concept classifier (Jahanian et al., 2020; Wang et al., 2021; Chen et al., 2016) to identify trajectories in the latent space that steer the concept, which resembles identifying a concept operator through

Output Optimization. Similar to RepE methods using Unsupervised Feature Learning (see Section 4.3), Voynov & Babenko (2020) discover directions corresponding to changes in a concept by having humans interpret the effect of the changes. Dong et al. (2024a) take inspiration from RepE for LLM safety to derive steering vectors that can be projected out to increase the safety of generated images.

### 10.8.2 Games

RepE has been used to interpret the internal world models learned by Transformer models that were trained in an autoregressive fashion on board games. Nanda et al. (2023) find that Othello-GPT represents the color of tiles linearly, and steering that vector affects which moves the model believes to be legal or not. Karvonen (2024) conduct similar experiments for a Chess-GPT model, where they find that individual pieces and the skill level of players are linearly represented and can be steered. These works indicate that autoregressive Transformers can learn to implicitly represent world models. Mini et al. (2023) identify a vector that successfully steers the goal pursued by a policy trained to solve mazes. This is promising since it could allow us to directly identify and alter the goals pursued by capable AI Agents.

---

**Takeaway: Applications**

RepE has been used in

- **AI Safety** to reduce harmfulness, increase truthfulness, and control refusal behavior.
- **Interpretability** to confirm the presence and shape of concept representations.
- **AI Ethics** to reduce societal bias, align LLM behavior with human preferences, and control ethical reasoning.
- **Knowledge Editing** to change or unlearn knowledge and adapt model encoded beliefs.
- Other areas such as controlling task execution and text generation and improving general model performance.

---

## 11 Comparing RepE to Other Methods

Table 12: Difference and Similarities of related Families of Methods to RepE

| Other Method | Difference to RepE |
|---|---|
| Prompting | Changes inputs instead of activations or weights |
| Soft-prompting | Operates on token embeddings instead of hidden states |
| Fine-tuning | Doesn't target specific concept representations |
| Decoding-based Methods | Operates on logits instead of hidden states |
| Mechanistic Interpretability | Bottom-up vs Top-Down view |
| Activation Patching | Fully replaces activations instead of adapting them |
| Probing | Only does Representation Identification and not Control |

### 11.1 Related Methods

There is a range of related methods that attempt to identify how concepts are represented in Neural Networks or that aim to control the behavior of LLMs.

**Prompting.** Changing the (system-)prompt given to an LLM is the most straightforward and commonly used way to steer the behavior of a model. However, it does so without any changes to weights or activations and does not provide any interpretability benefits. Furthermore, RepE does not take up space in the context window, while the additional tokens from prompting, especially when using many in-context examples, increases the computational cost.

**Soft-Prompting.** Similar to RepE, soft-prompting modifies the model in a continuous embedding space. It does so by operating on the input embeddings, whereas RepE modifies internal activations or weights. However, soft-prompting does not have the goal of identifying the representation of a concept.

**Fine-tuning.** Fine-tuning modifies the weights of a pre-trained model by further training it on a specific dataset or reward signal. While fine-tuning does influence the model's representations, it does not do so in a manner that is targeted to identifying and controlling representations of specific concepts. Instead, it applies some uninterpretable, non-sparse changes to the weights that lead to the desired outputs. Unlike RepE, fine-tuning does not allow for precise and isolated control of concepts of interest. Furthermore, many RepE methods provide continuous control, where a concept can be increased or decreased to the desired intensity. As mentioned in Section 6.2.1, there is an overlap between fine-tuning and RepE methods that do output optimization and apply a weight steering function.

**Decoding-based methods.** Decoding-based approaches steer the behavior of LLMs by shifting the token probabilities during the decoding step. Contrastive Decoding takes two sets of token probabilities, contrasts them against each other, and consequently shifts the token probabilities (Li et al., 2023b). Often, this means taking predictions from two different models, one of which is better at the desired behavior. Alternatively, contrastive predictions can be taken from the same model on contrastive inputs. Similar to RepE methods, this method steers the behavior of the model during inference time. However, contrastive decoding requires multiple forward passes per generated token. Furthermore, it only superficially shifts token probabilities and does not tap into the internal representations of LLMs.

**Mechanistic Interpretability (MI).** MI is not a specific method, but a field of study that aims to reverse engineer the mechanisms learned by Neural Network into human-understandable algorithms. Although RepE partially grew out of MI, Zou et al. (2023a) present it as a counterproposal. MI takes a bottom-up view by studying individual neurons and their connections into circuits. While such approaches can explain simple mechanisms well, they struggle to scale to higher-level concepts and processes. In contrast, RepE takes a top-down view by interpreting how high-level concepts are implemented as patterns in the representational space spanned by a large population of neurons. However, in practice, there is no clear delineation between these approaches.

**Activation Patching (AP).** In AP, the activations for some neurons on a specific input are replaced with different activations derived from other inputs. By observing the counterfactual outputs, researchers can study the causal relationship between internal representations and outputs. In contrast, RepE modifies the existing activations instead of fully replacing them. RepE also focuses less on a causal understanding of individual neurons and more on usefully steering behavior.

**Probing.** In Probing, a classifier is trained to predict the occurrence of a concept in the model's activations. Classically, probes are trained to interpret Neural Network. Furthermore, they can be used to detect concepts like task drift (Abdelnabi et al., 2024a) or deceptive reasoning (MacDiarmid et al., 2024; Goldowsky-Dill et al., 2025) during inference such that a more expensive safety procedure can be triggered. However, they can also be used as a method for Representation Identification in RepE or as a way to dynamically modulate the strength of the steering function.

## 11.2 Meta-study comparing to Other Methods For Behavior Control

Other methods for controlling the behavior of LLMs include prompting, fine-tuning, and decoding-based methods. We describe conduct a meta-survey that compares them to RepE according to their effectiveness and impacts on the model's capabilities.

For every paper, our meta-survey compares whether the proposed RepE approach or the compared approach works better for each paper. If available, we also compare which approach retains higher general capabilities and how many samples were used. When an experiment in a paper was run for multiple tasks, we averaged the results over the tasks. When there are multiple compared methods from the same category of approach, we compare to the best-performing one. Lastly, we want to caution the reader that most available comparisons come from RepE papers, thus, biasing the results in favor of RepE.

Table 13: We compare RepE to prompting, fine-tuning, and decoding by the amount of papers where it is better, equal, or worse at controlling a concept or retaining capabilities.

| RepE vs | Prompting | | Fine-tuning | | Decoding | |
|---|---|---|---|---|---|---|
| | Control Effectiveness | Capability Retention | Control Effectiveness | Capability Retention | Control Effectiveness | Capability Retention |
| better | 18 | 4 | 18 | 8 | 5 | 3 |
| equal | 0 | 1 | 3 | 2 | 0 | 0 |
| worse | 7 | 2 | 5 | 1 | 2 | 0 |

A list of all papers from this metastudy is provided in D and Table 14 and Table 15 respectively show experimental results for each comparison of RepE with prompting and fine-tuning respectively.

### 11.2.1 Prompting

There are 24 papers (listed in Appendix D) that provide experiments comparing the effectiveness between RepE and a prompting method. Across the surveyed papers, RepE is more effective than prompting at steering the target concept in 75% of the cases (see Table 13). Furthermore, it often had less impact on the general capabilities of the model. Notably, AxBench (Wu et al., 2025) conducts a thorough comparison between multiple RepE methods and prompting and finds prompting to perform better at detecting and steering concrete content-related concepts.

### 11.2.2 Fine-tuning

There are 25 papers (listed in Appendix D) that provide experiments comparing the effectiveness between RepE and a fine-tuning method. Hereby RepE methods such as ReFT (Wu et al., 2024c) and BIPO Cao et al. (2024a) that are adjacent to fine-tuning are counted towards RepE. Table 13 shows that RepE steers the target concept more effectively in 72% of comparisons, while fine-tuning is only better in 20% of cases. A majority of comparisons find that fine-tuning deteriorates general model capabilities more heavily in most cases. Figure 3 lightly indicates that RepE might be more sample effective, outperforming fine-tuning in every comparison under 500 samples.

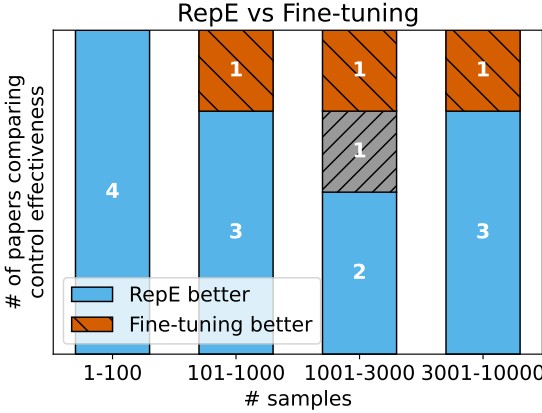

Figure 3: The number of papers using a number of samples where RepE or fine-tuning offer more effective control.

### 11.2.3 Decoding-based Methods

There are 7 papers (listed in Appendix D) that provide experiments comparing the effectiveness between RepE and a decoding-based method. RepE-based methods outperform decoding-based methods in 71% of cases and have much less impact on general model capabilities.

### 11.2.4 Combining Methods

One great advantage of RepE is that it works well in combination with other approaches. All four papers (listed in Appendix D) that evaluate this find that combining RepE with a prompting, fine-tuning, or decoding-based method delivers more effective control than any of them alone (Banerjee et al., 2024; Stickland et al., 2024; Li et al., 2023a; Wang et al., 2024c). This is typically tested in papers where RepE does not outperform the respective baseline method.

> **Takeaway: Comparing RepE to Other Methods**
> Other methods for identifying and editing concept representations, like Activation Patching or Concept Erasure, are less applicable for controlling output behaviors or have some overlap with RepE. Our meta-survey shows that RepE offers more effective control at a lower decrease of capabilities compared to prompting, fine-tuning, and decoding-based approaches.

## 12 Why Does It Work?

It is somewhat surprising that relatively simple operations on the representations of a model allow us to control high-level behavior while not destroying capabilities. While there are specific reasons for the effectiveness of different methods, this section offers some general explanations for this phenomenon.

**LLMs already represent human-understandable concepts.** LLMs seem to represent some human-understandable concepts like refusal or honesty. This makes it possible to tap into those representations and change their behavior. Thus, RepE does not need to learn the concept representations anew but simply identifies and promotes them.

**Models have structured representations.** LLMs represent concepts in a structured way. RepE can uncover and exploit this structure. For example, concepts might be represented as linear directions in the activation space of LLMs. This Linear Representation Hypothesis (LRH) (Park et al., 2024b) implies that scaling the direction corresponding to a concept increases the intensity of that concept. Many RepE methods are explicitly inspired by LRH or implicitly built on it. If the LRH is true, it explains the success of many RepE methods.

The plausibility of the LRH has been an area of much discussion in the interpretability community. In fact, some features have been proven to be represented linearly in autoregressive models trained for board games (Nanda et al., 2023; Karvonen, 2024) and in LLMs. Many concepts in LLMs have been decoded through linear probes (Li et al., 2024e; Abdou et al., 2021; Grand et al., 2018), and the success of many Representation Engineering methods is itself evidence for linear representations of high-level concepts. On the other hand, Engels et al. (2025) find non-linear features, such as circular representations for the days of the week.

**Resilience to activation manipulation.** Neural Networks are able to remain functional after their activations are manipulated. For example, adding small random perturbations to the activations does not strongly affect model performance (Reagen et al., 2018). This resilience might be attributable to dropout during training. Furthermore, the concept operator is derived from previous activations of the same network, which could mean that the manipulated activations still remain in the usual distribution of activations. This makes it possible to retain general capabilities.

**Control over intermediate variables.** Neural Networks can be seen as programs where activations are intermediate memory containing variables and are functions that perform operations on these variables. Changing variables in that memory changes the further computation and, ultimately, the output. When

RepE operates on representations of concepts in the activations, this is similar to editing semantically meaningful variables in the intermediate memory of a Neural Network.

> **Takeaway: Why it works**
> Models represent human-understandable concepts in structured, possibly linear ways. By manipulating these latent variables, we can influence the model's later computations. LLMs remain functional since they are somewhat resilient to internal manipulations.

## 13 Challenges in Representation Engineering

### 13.1 Empirical Weaknesses

**Multi-Concept Control.** Applying RepE for multiple concepts at once reduces the effectiveness of steering (van der Weij et al., 2024; Scalena et al., 2024) and leads to larger reductions in model capabilities (Scalena et al., 2024). This might be because applying multiple edits can lead to interference between concept representations (van der Weij et al., 2024). An exception is that combining specific vectors derived by BiPO Cao et al. (2024a) seems to increase their effectiveness. Improving multi-concept control is recognised as one of the major challenges in RepE (Cao et al., 2024a; Postmus & Abreu, 2024; van der Weij et al., 2024; Dong et al., 2024b; Chu et al., 2024). Multi-property steering is important since model providers will want to steer for more than one concept at a time. For example when van der Weij et al. (2024) combine steering vectors for Myopia, Wealth Seeking, Sycophancy, Agreeableness and Anti-Immigration together, the steering effect for each concept is reduced.

While there are proposals to steer different concepts in different layers (van der Weij et al., 2024), this limits the number of concepts that can be steered simultaneously. Furthermore, Scalena et al. (2024) dynamically adjust the strength of interventions which reduces the negative effects of multi-concept control. Further, there are attempts to control broader representations, like "human preferences", that encapsulate multiple desired concepts Liu et al. (2023). Nguyen et al. (2025) improve multi-concept steering by sparsely applying each steering vector only at some tokens and enforcing orthogonality between steering vectors to reduce interference. Lastly, Beaglehole et al. (2025) report progress on multi-concept steering based on non-linear feature learning.

**Long-Form Generation.** While RepE can effectively control short answers, there is a lack of evidence for its ability to control long-form generations and multi-turn conversations. It would be problematic if RepE would lose it's ability to control LLM behavior when a long text is generated or the conversation continues for multiple turns. Indeed, there is preliminary evidence that operators identified from a training set with short answers do not effectively steer long generations (Pres et al., 2024). This calls for a thorough evaluation of new RepE methods for longer generations. Recent work has started to address this problem by retaining the natural distribution of activations (Cao et al., 2024a) and dynamically adjusting the representation control throughout a generation (Scalena et al., 2024).

**Out-of-Distribution Generalization.** To effectively steer a concept, our concept operator should generalize from its training data to different contexts, prompt templates, or different generation tasks. For example a concept operator for harmfulness might only be derived on english inputs and fail to prevent harmful responses in other languages. Additionally, if a concept operator does not generalize across different situations, we cannot argue that we have found a general representation of the concept. However, this still presents a challenge for current RepE pipelines, which become less effective when the system prompt differs between the training and test set (Tan et al., 2024) and when the operator is identified *after* the answer but employed *before* the answer (Simhi et al., 2024). Evaluating and improving the ability of RepE methods for OOD generalization will be crucial (Mack & Turner, 2024a; Zou et al., 2023a; Li et al., 2023a; Ackerman, 2024).

**Deterioration of Capabilities.** RepE generally leads to a reduction in general language modeling capabilities and the quality and diversity of generated text (Scalena et al., 2024; Stickland et al., 2024; Zhang et al., 2024c). Aside from the quality of the generated text, Park et al. (2024a) points out that steered models

can be worse at instruction following. While this reduction is rather small, it still represents an important cost that might stop model providers from using RepE. Increasing the strength of the intervention reduces the capabilities (Wu et al., 2025; Durmus et al., 2024). This effect could appear because RepE disrupts the underlying structure of a model's activations (Zhang et al., 2024c). Thus, finding improvements to reduce or better navigate the trade-off between control effectiveness and general model capabilities will be crucial (Li et al., 2024b; Wu et al., 2024b; Wang et al., 2024c). Efforts to solve this problem include fine-tuning the model to prevent reductions in quality through RepE (Stickland et al., 2024), moderating the intervention strength to only steer when necessary (Li et al., 2024d), and more precisely localising the concept representations (Wang et al., 2024b). Others attempt to mitigate this by retaining the underlying structure of the LLM (Zhang et al., 2024c), keeping the magnitude of activations consistent (Pham & Nguyen, 2024), or providing an output-based goal to retain text quality (Xu et al., 2024b).

**Learning Specific Concept Representations.** Ideally, RepE would steer a concept without influencing other, unrelated concepts. However, in practice, RepE methods struggle to isolate a concept representation from other concepts. For example, steering towards happiness reduces refusal rates for harmful requests (Zou et al., 2023a), steering away from harmfulness makes the model less helpful (Wolf et al., 2024), and steering gender awareness can increase age-based biases (Durmus et al., 2024). A failure to isolate a specific concept representation might cause undesired side-effects that make RepE less usable in practice.

**Learning Complete Concept Representations.** We would like RepE to identify a concept operator which represents the concepts with all its aspects and in different contexts. A concept like Honesty is multi-faceted and is applied differently depending on the context. Failing to identify a complete representation of honesty means some aspects of the concept are not improved and the steering might fail in some contexts. However, identifying such complex representations is challenging and likely requires a large coverage of aspects and contexts in the training data, as well as a concept operator that is expressive enough. Chen et al. (2024b) address this by learning multiple orthogonal steering vectors that are supposed to capture different aspects of a concept.

**Unreliability.** RepE methods are generally unreliable and tend to fail in three ways. Firstly, they are sensitive to the hyperparameters chosen. Often, a small seemingly insignificant change in hyperparameters, such as small changes to the steering strength, can cause significant performance decreases (Zhang et al., 2024c; Adila et al., 2024b). This indicates that the chosen RepE setup is not robust. Secondly, there are often concepts that cannot be successfully steered. While it is common for papers to mainly report on the successful applications of their method, more neutral investigations often find concepts, like reducing narcissistic behavior, that can not be successfully steered (Tan et al., 2024; Tanneru et al., 2024). Braun et al. (2025) provide evidence that some concepts are not effectively steered because the difference between positive and negative activations is not consistent and not linearly separable. Lastly, even for interventions that are successful in steering the average behavior, there are inputs for which the steering effect is negative (Tan et al., 2024; Stickland et al., 2024). This can especially be the case for examples that were already positive with respect to the concept (Im & Li, 2025). This unreliability hinders the deployment of RepE methods.

**Unreliable Interpretability.** RepE is often used as a tool to interpret LLM representations, but it is questionable whether they provide strong evidence or are largely misleading. Firstly, a failure to identify an effective linear operator for a concept does not prove that the concept is not linearly represented in the model. It could merely be a failure of the method to find the relevant directions. Secondly, it has been shown that finding an effective intervention in specific layers or attention heads does not prove that the concept is actually localized there (Wang & Veitch, 2024). Thirdly, there can be multiple different vectors that successfully steer the concept. For example, (Goldman-Wetzler & Turner, 2024) find 800 orthogonal vectors that steer "write code". Consequently, a direction that steers the model is not necessarily the single direction that represents the concept (Subramani et al., 2022).

**Requires Access to Models' Internals.** RepE methods rely on access to models' internals and activations. This limits who is able to create and employ RepE methods. Xu et al. (2024b) partially circumvent this by using RepE on white-box models to find safety vectors, use them to derive attack prompts, and transfer those attack prompts to a black-box model.

**Computational Cost.** Although RepE is highly efficient during inference and cheap to train, multiple papers aim to improve the computational efficiency of deriving or employing RepE methods. This can be done by removing the need to perform expensive hyperparameter tuning (Xu et al., 2024b) or expensive iterative optimization (Qiu et al., 2024; Jorgensen et al., 2024). The additional operations necessary to manipulate activations during inference can be avoided by instead manipulating weights (Wang et al., 2024a).

## 13.2 Principled Challenges

In addition to these previous general challenges and weaknesses, we further break down the challenges according to each step in the pipeline.

### 13.2.1 Challenges in Representation Identification

**Spuriously Correlated Concepts.** Current RI methods are mostly not able to disentangle concepts that are correlated. As a result, the identified concept operator will also capture and steer the spuriously correlated concept (Tan et al., 2024). Concepts can be correlated because they often occur together in the inputs, because the output scoring evaluates both highly or because their representations tend to be activated at the same time. For example, instructing a model to write faulty code can simultaneously trigger the representation to write faulty code and the representation for general unethical behavior (Betley et al., 2025; Soligo et al., 2025). Using RI techniques like difference-in-means or linear probes will consequently lead to a steering vector that captures the representation for faulty code and general unethical behavior. This presents a major hurdle for precisely identifying a concept representation. Sparse Autoencoders might be able to address this by identifying disentangling concept representations.

**Concept Misspecification.** Representation Identification forces us to specify the target concept by providing a scoring function or inputs related to the concept. Through these, we are hoping to elicit the model's representation of this concept. However, if the concept specification differs from the intended concepts, the identified concept operator will not be accurate. For example, in Output Optimization we might use an LLM to score how honest outputs are to learn a concept operator. But if the LLM-judge fails to detect some forms of dishonesty, the resulting concept operator will also fail to steer these forms of dishonesty.

Furthermore, even when the specified inputs or scoring functions are accurate, they might activate other representations than the ones for the desired concept. For example, we might instruct a model to be honest or dishonest and hope this activates the model representations of "I will be honest/dishonest". However, for a model that was trying to deceive, such an input might instead activate a representation for "The human wants me to be honest/dishonest", which is importantly different from the intended concept. Deng et al. (2025) point out that RI methods based on contrastive pre-prompts implicitly assume that the model actually follows the pre-prompt.

Additionally, models might use concepts that are not understandable for humans. They might use an ontology that differs from ours or discover new concepts. In this case, it would be difficult to specify and steer these concepts.

**Interference from Superposition.** Interpretability research has shown that LLMs are polysemantic since they represent more features than they have dimensions (Elhage et al., 2022). Thus, networks represent features in superposition, meaning that features are not all orthogonal to each other, which results in interference between features. In practice, this will mean that controlling a concept representation will also steer some other concepts. Thus, concept operators are not specific to a concept and have side effects on unrelated concepts. For example steering a model towards academic citations could accidentally also steer it to writing HTTP requests if there is interference between these representations Bricken et al. (2023). Nguyen et al. (2025) attempt to reduce interference by enforcing orthogonality between concept vectors, thus improving multi-concept steering.

**Assumptions on Available Data.** RepE methods often only work when specific data is available. Firstly, it is not always possible to provide contrastive inputs that describe the behavior and its opposite (Jorgensen et al., 2024; Cai et al., 2024; Postmus & Abreu, 2024). For example using contrastive inputs might not be the right methods for non-binary concepts like the days of the week. Secondly, many methods rely on

expensive human-annotated ground-truth data (Adila et al., 2024b;a). For example steering against societal biases might involve humans annotating or writing biased and unbiased texts, which can be prohibitively expensive. Lastly, methods are not designed to handle noisy and biased data Adila et al. (2024a).

**Reliance on Models' Own Representations.** RepE methods tap into the existing representations of a model. Consequently, a concept cannot be controlled with RepE if the network did not learn a representation for it. If a model never learned a representation for being honest, RI methods will not be able to find an effective concept operator to make the model more honest.

**Interpretability Lacks Ground Truth.** The challenge of developing methods that find representations related to a concept is very difficult because there is no ground truth for where and how the model represents a concept. Thus, we cannot know whether we have identified an accurate representation or what the semantic significance of the concept operator is (Herrmann & Levinstein, 2024). While we might find a vector that causes the model to be more honest and detect dishonest generations, we have no guarantee that this maps to how the model actually represents honesty.

### 13.2.2 Challenges in Representation Operationalization

**Assumptions About Models' Representations.** Methods for identifying concept operators come with assumptions about the shape and geometry of representations in the models' activations. Common assumptions about representations are that concepts are represented as linear direction (Qiu et al., 2024), that representations do not change throughout a sequence of inputs, that they are the same across context, that they are localized in a single layer, and that they do not rely on interactions between layers.

However, if the true form of the representation does not match these assumptions, it will lead to unsuccessful or less effective Representation Identification and Control. If a concept representation is non-linear, a linear concept operator can at most be a first-order approximation of the true representation and thus provides less precise and effective steering. Similarly, when the true representation of a concept depends on the context or can change throughout trajectories, steering with one static concept operator is at best optimal in a subset of situations. Additionally, when concepts rely on interactions between layers, single layer steering cannot capture all aspects of that concept or could fail to correctly influence the concept in later layers.

These assumptions have been criticized (Luo et al., 2024; Chu et al., 2024). See also our discussion of the validity of the Linear Representation Hypothesis in Section 5.1. Assumptions have been changed and weakened by shifting from a point-in-space to a direction-magnitude view (Pham & Nguyen, 2024), adapting the strength and direction of steering to the context (Wang et al., 2024c), or applying non-linear operations (Qiu et al., 2024; Hoscilowicz et al., 2024).

An additional challenge is that a feature might not have a single representation. For example, (Goldman-Wetzler & Turner, 2024) find 800 orthogonal vectors that steer "write code" and (Mack & Turner, 2024b) finds 200 independent vectors with which a model represents that a request is harmless. While this might be due to weaknesses in the specific methods or because of wrong assumptions about the representations geometry, it could also mean that some concepts are represented in multiple ways inside a model. This challenges traditional notions in RepE, where it is assumed that there is one representation of the concept.

### 13.2.3 Challenges in Representation Control

**Shifting Activations off Their Natural Distribution.** Manipulating the weights and activations of a model can shift the representations off of their natural distribution. In turn, this can lead to a deterioration of LLMs' capabilities since later computations in the model might be disturbed by the unnatural shift in activations. While steering was found to only cause small shifts in the distribution (van der Weij et al., 2024), this effect could be detrimental when the strength of intervention is increased or many concepts are steered at the same time.

> **Takeaway: Challenges in Representation Engineering**
> RepE struggles with steering multiple concepts at once, controlling long-form generations, and providing

a reliable steering effect. These weaknesses arise because of challenges like the inability to disentangle spuriously correlated concepts, to posit correct assumptions about models' representations, and to retain the models' natural distribution of activations.

## 14 Opportunities for Future Research

### 14.1 Opportunities to Improve Representation Identification

**Refining Identified Operators.** The identified concept operator could be improved to make it better suited for control. For example, an operator identified through Input Reading could be refined by fine-tuning it to produce desired outputs or by decomposing it with an SAE and removing unrelated features. In general, there is promise in developing pipelines that combine different RI methods. For example, Wu et al. (2025) suggest that finding concept operators by jointly learning for concept detection and steering can be more effective.

**Data-centric RepE.** Most efforts for improving RI focus on new calculations for deriving the concept operator. However, recent work indicates that the quality of a concept operator is strongly dependent on the quality of the dataset it is trained on (Tan et al., 2024; Braun et al., 2025) and improves when it is trained on more data (see Appendix A.2). Akin to data-centric approaches to train better models (Zha et al., 2025), a focus on enhancing the quality and quantity of data could be key to improving RI. Furthermore, detailed investigations of the properties of datasets that lead to strong steering performance could unlock important insights.

**Better Methods for Scoring Outputs.** Currently, most Output Optimization methods simply compare outputs to a ground-truth text. There are more accurate and flexible ways for scoring the alignment of outputs with a concept. One could use a human judge, LLM-as-a-judge, a trained reward model, or specific context-dependent metrics. Furthermore, one could use the performance in an environment, like rewarding actions in a multi-agent negotiation setup based on the final agreement (Abdelnabi et al., 2024b).

**Combine with Other Interpretability Methods.** As described in Section 11.1, there exist other methods that aim to identify representations of concepts in LLMs. While these might not produce suitable concept operators themselves, they could provide information about the representation that can be leveraged for Representation Identification. At least, it would be interesting to compare the identified representations from different methods for the same concept.

**Finding and Analyzing Failure Modes.** We hypothesize that spuriously correlations between concepts and concept misspecification pose challenges for effective Representation Identification. However, there is no empirical evidence for this yet. Exploring such theoretically plausible failure modes could uncover new weaknesses to be addressed by improved RI methods.

**Automated Interpretability.** LLMs can judge which concepts a feature corresponds to by looking at highly activating inputs and outputs. This is commonly done in Unsupervised Feature Learning to identify concept operators. A similar approach could be used to refine concept operators identified by other RI methods. An LLM can automatically judge the inputs and outputs activating for multiple concept operators.

### 14.2 Opportunities to Improve Representation Operationalization

**Expanding on Linear Representations.** Most RepE pipelines assume that concepts are represented as linear directions. However, in practice, representations might be non-linear. As noted previously, there is early work on RepE for non-linear steering, but more exploration of non-linear representations is necessary.

**Context-Dependent Control.** Current RepE methods assume that the representation of a concept is the same for every context and thus apply the same intervention independent of the inputs. However, the model might represent a concept differently in different contexts. There is already work that dynamically adjusts the strength or composition of concept operators. However, future work could learn separate concept operators for a range of situations or train a small model that can predict the right concept operator to apply in a new situation.

**Modeling Trajectories of Representations.** Current RepE methods assume that representations of a concept remain static throughout a generation. However, it could also be that the representation of a concept changes over time. In that case, representations of interests are better modeled as trajectories of activations over inference steps.

**Inter-Layer Dependencies.** Recent work (Lindsey et al., 2024) shows that representations of some concepts can be spread over multiple layers. Current RepE methods are not capable of capturing these inter-layer dependencies. To address this, RI methods could concurrently optimize multiple concept operators at different layers.

**Modeling Interactions Between Concept Representations.** Current RepE methods aim to find an isolated representation of a single concept. But this misses any interactions and relationships between multiple concepts. For example, there could be a specific representation $o^{A \wedge B}$ if concepts $A$ and $B$ are both present, or there could be conditional representations $o^{B|A}$ of $B$ given that $A$ is present. Thus, attempting to identify interacting representations between concepts could uncover more rich concepts the model is using.

**More Complex Operators.** Current operators are mostly relatively simple. Perhaps more complex operators could capture more comprehensive and nuanced representations. It might be possible to train a Neural Network to predict a concept operator for the given inputs. Furthermore, an ensemble of concept operators could be trained on different portions of the data or different clusters of activations to capture different aspects of a concept.

**Feature Geometry as a Hyperparameter.** Current work starts by assuming the geometry of the concept representation and then aims to identify the concept representation adhering to that geometry. By trying out multiple geometries for one concept and then measuring their effectiveness, we can expect to find the geometry that most suits the specific concept representation. Additionally, RepE would benefit from further understanding the geometry of concepts in Transformer Language Models, which is already an active field of study (Jiang et al., 2024a; Csordás et al., 2024).

## 14.3   Opportunities to Improve Representation Control

**Combining Weight and Activation Steering Functions.** Representations are an interplay between weights and activations. Thus, it could be beneficial to develop methods that control weights and activations together. Naively, one could apply RepE for a concept to the weights and also apply it to the activations. Furthermore, it could be interesting to strengthen or weaken certain representations while training the model, thus guiding which representations are learned by the model.

**Complex Steering Sequences.** By applying steering for different sequences at different time points in a generation, RepE can be used to steer more complex behaviors or ensure that LLMs abide by a predefined pattern. For example, the providers of an LLM might want it to answer code-related questions by first steering for "detailed reasoning" while the problem is being analyzed, then for "not containing errors" while new code suggestions are generated, and lastly for "friendliness" while the model asks for clarifications.

**More expressive steering functions.** Krasheninnikov & Krueger (2024) find in a toy setting that more expressive steering functions, whose operators have more parameters, can reach a higher performance ceiling and control more complex tasks while requiring more data. This indicates there is promise in developing more expressive steering functions to push the Pareto frontier of steering effectiveness and data-efficiency.

**Controlling Learning Processes with RepE.** Yu et al. (2024) repeatedly ablate the refusal feature to learn more robust refusal mechanisms. This highlights that steering during training can be used to influence the representations that are being learned by the model. Aside from ablating features, representations can be positively steered potentially to make them more salient in the learning process. On the other hand, learning representations in specific ways can provide concept operators to steer model behavior. For example, Cloud et al. (2024) use Gradient Routing to control where in the network specific capabilities are localized, thus enabling to unlearn specific capabilities during inference.

### 14.4 Possible Applications of Representation Engineering

After outlining current applications in Section 10, we now suggest some further problems that could be addressed with RepE.

**Agent Goals.** RepE could be used to control the goals pursued by an LLM Agent. Previous work achieved this for RL agents (Mini et al., 2023), but the growing popularity of LLM agents trained with Reinforcement Learning (OpenAI, 2024) makes it timely to apply RepE to steer their goals. Controlling an agent away from dangerous and towards desirable goals could be a simple but effective technique for mitigating misalignment in LLM Agents. RepE might be uniquely capable of adjusting agent goals compared to prompting. An AI system that intrinsically values a goal is instrumentally incentivized to retain that goal (Bostrom, 2012), even when it is prompted to change its goal. However, RepE might be able to directly edit the goal the AI is pursuing by steering its own representations.

**Studying In-Context Learning.** Finding out how and why in-context learning (ICL) works is a hotly debated topic. By controlling representations, it might be possible to steer the hypothesis class ICL uses and thus make progress on understanding the patterns and algorithm enabling ICL.

**Preventing Deception.** AI systems behaving as if they are aligned with human interests while secretly pursuing different goals is a key worry in the AI Safety community. RepE could provide unique control over models' behavior when the AI is attempting to deceive the user (Park et al., 2024c). In this case, it will not suffice to instruct the model to be truthful. However, by directly accessing the model's own representations and thus influencing its "thought process", RepE might be able to control it to be more honest.

**Controlling Coooperation.** As LLM agents become more ubiquitous, they will often interact and negotiate with other LLM agents. Through RepE, it might be possible to control how agents behave and cooperate in multi-agent scenarios.

**Value Engineering.** RepE could be used to identify specific ethical values to let users personalize their LLM agent. For example, a user could specify which beings the AI should give moral consideration to, which moral theory it should apply, or what its stance on specific issues should be.

**Red Teaming.** RepE could be used when red-teaming and evaluating LLMs. Firstly, for open-weight models, which can easily be controlled towards harmful outcomes through RepE, it should be standard to also apply RepE during red-teaming. Secondly, for closed-weight models, applying RepE to jailbreak a model could serve as a worst-case safety failure that can be evaluated (Che et al., 2024).

**Studying the Development of Concept Representations.** RepE could be used to identify and steer representations for the same concept in different stages of training or different models, thus giving insights about when representations of a concept emerge and how they change. One could use RepE to study how concepts evolve throughout pre-training and how they are influenced by various post-training methods or how concepts emerge across models of different sizes in the same family.

### 14.5 Building a More Rigorous Science of Representation Engineering

The field of Representation Engineering is young and does not yet stand on a strong scientific foundation. To enable progress in the field, we point out the opportunity to make research into RepE more rigorous and to provide a better grounding for the field. The most important development here would be the introduction of a high-quality, widely accepted benchmark for measuring RepE methods, which we propose in Section 8.3 as well as the adoption of rigorous evaluation standards discussed in Section 8.2.

**Developing Theoretical Frameworks.** The field of RepE is almost entirely based on experimental evidence and lacks a clear theoretical framework. Fields such as Adversarial ML and Machine Unlearning have greatly benefited from a shared theoretical framework that formulates the settings and objectives (Cao & Yang, 2015). Such a theoretical framework could clearly state the problem RepE attempts to solve and provide criteria for success. For example, current theoretical work on RepE has found that difference-in-means is optimal at identifying a vector that shifts negative to positive examples (Im & Li, 2025; Singh et al., 2024).

**Causality and RepE.** RepE aims to identify representations that are causally related to a concept of interest and apply interventions that cause predictable changes in output behavior. Taking a causal perspective on RepE, the treatment could be the relation of a string to the concept or the intervention applied to the representations, and the effect could be the change in activations or in output behavior. Frameworks from causality, such as Structural Causal Models (SCMs) based on Directed Acyclic Graphs (DAGs) (Pearl, 2014) and the Potential Outcomes (PO) framework (Rubin, 1974) could provide a theoretical grounding for RepE.

A causal graph describing the computations of the model allows for the application of the do-calculus. This would enable us (1) to find minimal interventions that have the desired steering effect via the back door criterion, (2) to estimate the causal effect of representations despite the presence of unobserved confounders via the front door rule and (3) to distinguish interventions that have a direct or indirect effect on the desired outputs. However, this comes with multiple assumptions and requires one to identify how patterns in the activations of a network relate to nodes in the causal graph. Indeed, such an attempt has been made by formalizing (Geiger et al., 2024a) and conducting (Geiger et al., 2024b) the search for a DAG of high-level concepts that aligns with the neuron-level activity in the network.

Alternatively, using the PO framework does not require DAG and instead leverages covariates. This allows the use of tools such as *propensity score matching* or *doubly robust estimation* to assess the causal effect of modifications to representation and thus enables causal adjustments of the interventions. However, this assumes that confounding factors can be adequately controlled using observed covariates.

Lastly, causality can inform the design of input datasets from which concept operators are derived Deng et al. (2025). This can help to better isolate the causal mechanism from confounders and improve ood generalization.

**Studying RepE on Toy Setting.** Krasheninnikov & Krueger (2024) develop a simple classification task as a toy setting to run controlled experiments evaluating RepE methods for tasks of different complexities. Such toy settings can help us to gain a deeper understanding how and why RepE works. They can also serve as testbeds to evaluate the strengths and weaknesses of different RepE methods. Such toy setups could be used to study the impact of model size, data quality and assumed feature geometries.

**Building a Library of Concept Representations.** An ambitious effort could aim to map out a library of concepts and their representation in a specific model. This could serve as a valuable resource for later study. For example, by using this database, it might be possible to train models that can predict the concept operator for new concepts. Furthermore, such a database could serve as a baseline for the development of new RepE methods.

**Finding Best Practices.** Practitioners using RepE want to easily know which methodology is most effective. However, currently, it is not clear which method should be used practically. Finding best practices for using RepE and then describing them in an easily accessible form could greatly increase the adoption of this technique.

**Developing Tooling for RepE.** Great software tools for fine-tuning, such as the Transformers library (Wolf et al., 2020), make it easy to use this technique, thus boosting its adoption. Developing dedicated tooling, such as APIs or libraries, for RepE could be similarly valuable. An example of such work is Gemma Scope (Neuronpedia, 2025) which allows novices to analyze and steer SAE features derived from Gemma-2-2b.

---

**Takeaway: Opportunities for Future Research**

Future RepE methods can expand restrictive notions of representations, combine multiple RI methods and attempt more complex interventions. Untapped applications lie in steering goals and values of agents or preventing deceptive behavior. To progress the field of RepE, thorough evaluations and best practices are needed.

---

## 15    Conclusion

Representation Engineering (RepE) is a novel paradigm for controlling and interpreting Large Language Models by manipulating their internal representations. This survey has provided a comprehensive overview of RepE methods, focusing on different methods to identify, operationalize, and control the representations of the model. By synthesizing insights from over 100 recent studies, we have highlighted key advancements, methodologies, and challenges in this rapidly evolving field.

Our analysis reveals that most RepE methods identify representations by providing contrasting inputs and controlling them by adding a vector to the activations. Furthermore, many approaches are based on the Linear Representation Hypothesis. These methods have been applied to AI Safety, Ethics, and Interpretability by controlling concepts such as Harmfulness, Fairness, and Truthfulness.

We find that RepE tends to be more effective at a lower cost to generation quality compared to prompting, fine-tuning, and decoding-based methods. However, challenges remain in ensuring reliability, robustness, and quality, especially for tasks requiring long-form generation and multi-concept control. Issues such as spuriously related concepts, assumptions about feature geometries, and distributional shifts caused by RepE must be addressed. Further research should prioritize the development of more rigorous ways of evaluating and comparing RepE methods. Furthermore, researchers should explore representations that are non-linear, developing over time, spanning layers, and interacting with each other.

We believe RepE will prove to have important advantages compared to other control methods. RepE might be especially advantageous when there is a need to control not only the output of the model but also the internal processes by which it arrives at these outputs. As such, it is likely that RepE will become a common tool for engineers to control and for researchers to study LLMs.

### 15.1    Limitations

Our survey is only comprehensive with regard to RepE papers published before 31.8.2024. While we include many papers published until 31.2.2025 we do not comprehensively cover them all. Papers covered after this period are not covered.

While we attempted to synthesis the quantitative evidence on best practices for RepE in Section 7.2, we had to leave many questions unanswered. This is due to a lack of empirical comparisons that shed light on these questions.

### Broader Impacts

RepE is a general tool for controlling the behavior of LLMs. In many cases RepE has been devised and used for applications such as steering LLMs to reduce harmful outputs, align them with human intent, make them more honest or uncover societal biases. However, RepE faces a dual-use risk since these concepts can also be steered negatively. For example RepE has been used to jailbreak LLMs to bypass safety mechanisms and produce more harmful outputs. Similarly, RepE could be used to steer the model to become misaligned, generate deceptive text or to strengthen societal biases.

Additionally, RepE has been used to interpret model internals. However, it does not offer perfect explainability. Thus there is a risk that an illusion of interpretability is created where users trust the these explanations more than is warranted.

## Acknowledgments

The authors are grateful for thoughtful feedback from Ruta Binkyte, Dmitrii Krasheninnikov, Ivaxi Sheth, Sarath Sivaprasad, Alex Cloud and Simon C. Marshall.

This work was partially funded by ELSA – European Lighthouse on Secure and Safe AI funded by the European Union under grant agreement No. 101070617. This work was also partially supported by the ELLIOT Grant funded by the European Union under grant agreement No. 101214398. Views and opinions

expressed are however those of the authors only and do not necessarily reflect those of the European Union or European Commission. Neither the European Union nor the European Commission can be held responsible for them.

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

# A Meta-survey

In order to get a better feeling for how research on RepE is conducted, we conducted a meta-survey where we gathered which models and sample sizes were used. Furthermore, we collect statistics about the papers including authors and publication dates. This meta-survey only includes papers published before 31.8.2024.

## A.1 Commonly Used Models

The most popular models are the base and respective instruction-tuned models of Llama-2-7B, Llama-2-13B, Mistral-7B and Llama-3-8B.

We find that a majority of experiments (164 out of 229) are performed on base models that have not been trained with RLHF, while 65 experiments are conducted on preference-tuned models. While experimental results in Dong et al. (2024b); von Rütte et al. (2024) find a larger effect from applying RepE to base models than to instruction-tuned models, Bortoletto et al. (2024); Wang et al. (2024c) find no clear relationship between instruction-tuning and steering effectiveness.

Figure 4 shows that a majority of models to which RepE has been applied on lie between $3-10$ billion parameters. This is underlined by the fact that 9 out of the 10 most popular models for RepE are all between $7-13$ billion parameters. Furthermore, 8 of them belong to the Llama family of models or are derived from Llama models. This calls for more diversity in which model experiments are performed. However, the model scale might not be a crucial factor for the effectiveness of RepE. Across 4 papers that compare the steerability of models across different numbers of parameters within the same model family, none find a clear relationship between scale and steerability (Arditi et al., 2024; Xu et al., 2024a; Arora et al., 2024; Bortoletto et al., 2024). Furthermore, only Templeton et al. (2024) are able to experiment on a proprietary, state-of-the-art model, thus leaving open questions about how effective RepE is at larger scales and for the most capable models.

Almost all papers focus on transformer language models, while Paulo et al. (2024) find that CAA can also be applied to RNNs by steering the in-layer activations or the state-space. Furthermore, Adila et al. (2024b) apply RepE to a vision-language model. This indicates that RepE can be an architecture-agnostic method.

## A.2 Number of Samples

We note how many samples were used during Representation Identification to derive the operator. Figure 5 shows that most papers use between 100 and 1000 samples, showcasing that RepE can be relatively sample efficient. However, as with fine-tuning and ICL, RepE benefits from using more samples. Qiu et al. (2024), Yin et al. (2024), and Wang et al. (2024c) find that the effectiveness of steering improves as the number of samples for Representation Identification increases. However, Adila et al. (2024b) find no such relationship in their unsupervised method. Lastly, Krasheninnikov & Krueger (2024) find that more expressive steering functions, whose concept operators have more parameters and that contain more operations, work less well with a low amount of data but can better leverage high amounts of data.

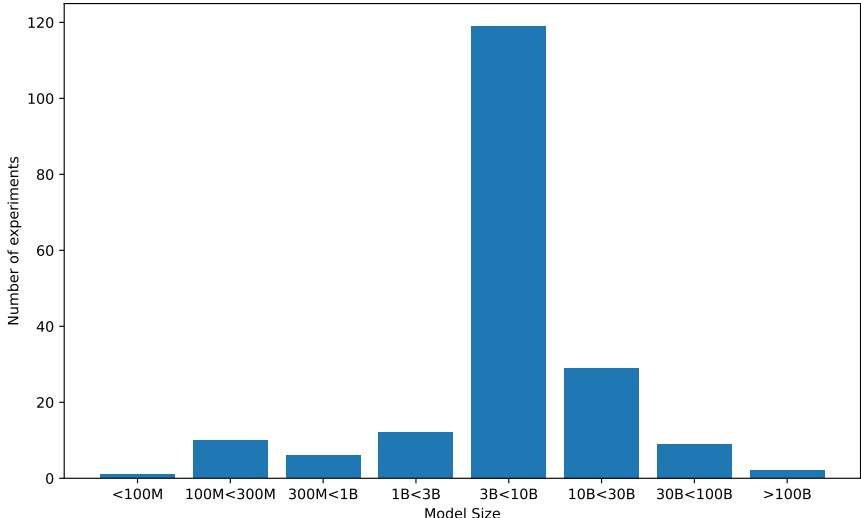

Figure 4: The number of experiments applying RepE to a model with specific numbers of parameters.

There is no general recommendation on the necessary number of samples, which depends on the model, target concept and method used. Building more sample-efficient methods and identifying the scaling laws of RepE are promising avenues of inquiry.

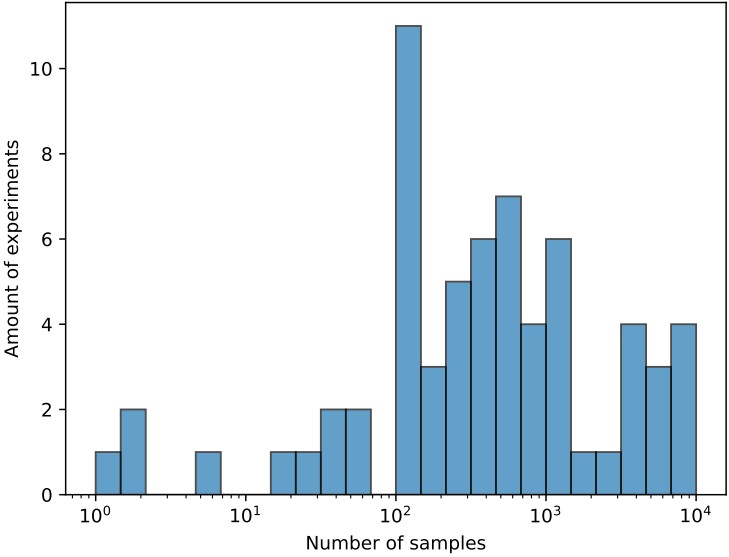

Figure 5: Amount of experiments that use a certain amount of samples.

### A.3 Statistics about Publications on Representation Engineering

**Applying vs. Improving.** A majority of papers set out to make methodological improvements to RepE (60 out of 87), while 22 mostly focus on applying RepE to solve a problem. Only 3 papers aim to evaluate RepE methods.

**Academic vs. Industry Papers.** Work on RepE is mostly done by academics, with 58 out of 87 papers only featuring academic co-authors. Another 19 papers are collaborations between academia and industry, while only 8 papers are solely authored by industry researchers.

**Publication Date.** We take the date of the initial publication of a paper and plot them in Figure 6. Here, we count the first time a paper was published. If a paper was first uploaded on arXiv and later published in a conference, we count the date of the initial arXiv submission.

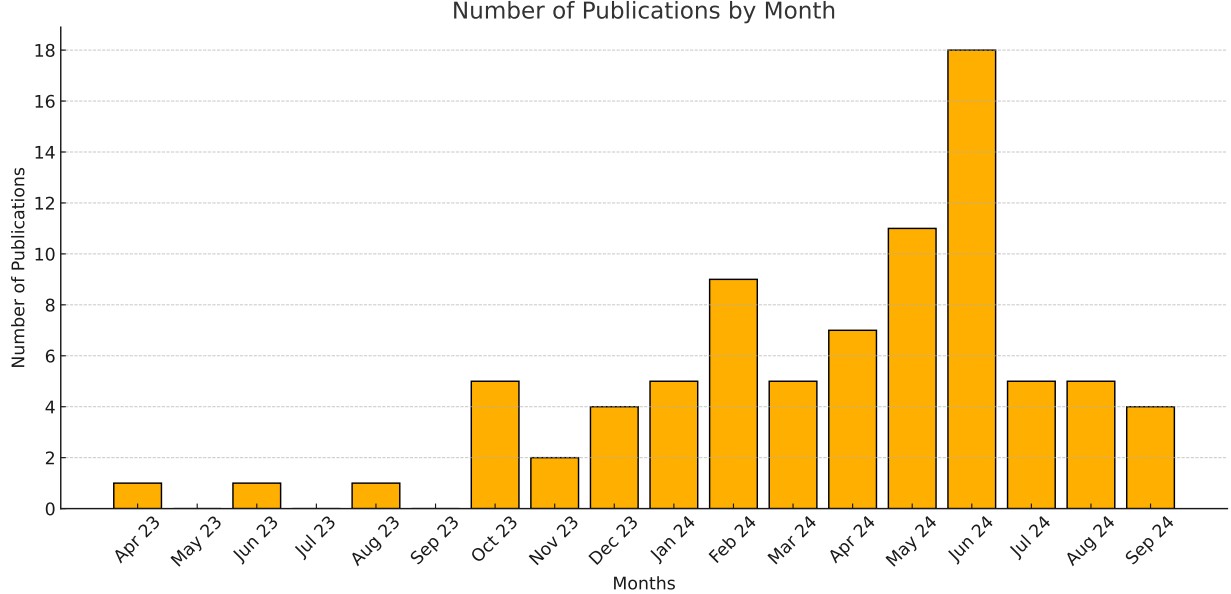

Figure 6: Linechart with publications over time.

---

**Takeaway: Meta-survey**

- **Models**: A majority of models used have between 3-10 billion parameters. According to current literature, there appears to be no relationship between RepE effectiveness and model scale.

- **Samples**: More samples make RepE more effective.

- **Publication statistics**: A majority of papers on RepE come from academics. The literature is very new but growing rapidly.

---

# B  Survey Methodology

We conducted a structured survey that resulted in a new taxonomy and summary of existing applications but also serves as a meta-study that compares RepE to other methods and gathers interesting meta-statistics.

## B.1  Literature Search

Firstly, we started with 3 seminal papers (Zou et al., 2023a; Turner et al., 2024; Li et al., 2023a) and conducted a forward search. Secondly, we used the combinations of search terms *{"Activation Steering", "Representation Engineering", "Concept Activation Vectors", "Linear Probe + Steering"} + {"Language Model", "LLM", "Neural Network"}*. Thirdly, the citations of relevant papers were checked.

To distinguish papers on RepE from related methods, we adopt five criteria for inclusion and point to papers that were excluded by this criterion.

- **Operates with LLMs.** This excludes the sizeable literature on using steering vectors in image generation models like GANs and Diffusion models, which we describe in Section 10.8.

- **Identifies a Concept Representation.** A representation related to the target concept needs to be identified instead of only training the network.

- **Controls Models' Behavior.** This excludes work that solely focuses on interpreting LLMs without modifying representations.

- **Performs Post-hoc Representation Identification.** We focus on identifying representation after training instead of retraining the network to produce desired concept representations.

- **Steers the Intermediate Representations.** We focus on representation steering instead of shifting token embeddings or logits.

This exhaustive literature search was conducted in September of 2024 and only covers papers published until 31.8.2024. We include many RepE papers published after that date but do not guarantee a complete coverage of later publications.

### B.2 Extracting Information from Papers

While surveying papers, we extracted key information upon which the survey was built. A full list of the information we extract can be found in Appendix C.

Regarding the methodology, we developed the taxonomy after a first non-exhaustive read of the literature. On a second, exhaustive read, we classified where methods fall in our taxonomy but also recorded many methodological details that allowed us to map the space of choices when applying RepE. Furthermore, we identify which target concepts were engineered and what problem the paper attempted to solve, thus letting us cluster papers into categories of applications. We noted any experimental comparisons with other RepE methods or other methods for controlling LLMs' behavior. Additionally, we gathered experimental details like the model, dataset, and sample size used. Furthermore, we extracted meta-information like the date and venue of publication and association of the authors. To gather views on RepE, we recorded citations where authors expressed how they see RepE as useful, how they think it works, what weaknesses they see, and which future work would be valuable. See Appendix C for all information we extracted from the papers.

We used the assistance of Claude-3.5-Sonnet to extract information from papers. While this sometimes sped up the process of identifying key information, it is not sufficiently reliable to be trusted to retrieve information faithfully. Thus, we read every paper ourselves and confirmed the correctness of any LLM-generated answers before including them in this survey.

## C Information Extracted from Papers

**General.**

1. Name of the method

2. Short summary of the paper

3. Does it improve Representation Engineering methods or apply it to a problem?

4. Which area and problem does it apply RepE to?

5. Open problems/future work

6. Publication month and year

7. Venue

8. Are the authors from academia, industry or a mix?

**Identification.**

1. Is it an application of a previous method? Which one?

2. Description of the identification method

3. Is the information about the concept obtained by reading from inputs, optimizing for outputs, or optimizing for an internal loss function?

4. How are concept-related activations elicited?

5. How is the concept vector identified from activations?

6. Is the method supervised or unsupervised?

7. What data was used for identification?

8. An example of the prompting format

9. At which layer, in which component, and for which token are activations read?

**Control.**

1. Description of the control method

2. How does it engineer activations?

3. Does it change activations or weights?

4. At which layer, in which component, and for which token does engineering take place?

5. Is the change in activations static or dynamically adjusted per input?

6. Aside from control, does the paper use the concept vector for detection?

**Concepts.**

1. Which concepts were identified and steered?

2. Which concepts were successfully steered?

3. Which concepts were not successfully steered?

**Experimental Details.**

1. Is there an experimental comparison to non-RepE methods? If yes, which method is superior?

2. Is there an experimental comparison to other RepE methods? If yes, which method is superior?

3. How many examples were used for identification?

4. Which datasets was the control method evaluated on?

5. Which models were experimented on?

6. Summary of interesting results

Furthermore, we noted **arguments related to RepE** that were only contained in a few papers:

1. How does RepE relate to other methods/areas?

2. Why does RepE work?

3. What is the theory/impact/promises/justification of RepE?

4. What weaknesses does RepE have?

5. What are principled advantages/disadvantages between RepE Methods?

Table 14: Collection of studies that compare the steering effect and model quality between RepE and Prompting methods

| Paper | RepE method | Prompting method | Concept | Evaluation Task | Steering Effect | | | Model Quality | | |
|---|---|---|---|---|---|---|---|---|---|---|
| | | | | | Metric | RepE | Prompt | Metric | RepE | Prompt |
| (Zhang et al., 2024c) | ARE | GCG Self-Reminder | Harmfulness Truthfulness | Jailbreaking Mitigating hallucinations | Refusal Rate ↓ Hallucination Rate ↓ | 0.17 56.1 | 18.9 40.1 | - | - | - |
| (Cai et al., 2024) | SELF-CONTROL | System Prompting | Emotions Toxicity Privacy Reasoning | Changing expressed emotion Reduding Toxicity Data Leakage Math Problems | Score ↓ Toxicity Score ↓ % Private data ↓ Accuracy ↑ | 2.40 0.17 0 37.3 | 2.10 0.27 77.5 40.0 | - | - | - |
| (Arditi et al., 2024) | ORTHO | GCG-T | Harmfulness | Jailbreaking | Attack Success Rate ↑ | 46.1 | 29.3 | - | - | - |
| (Zhang & Viteri, 2024) | CAA | CoT Prompted | Reasoning | Increase Performance | Accuracy ↑ | 71.5 | 69.8 | - | - | - |
| (Chen et al., 2024b) | TrFr | Few-shot Prompting | Truthfulness | truthful multiple-choice | True*Info ↑ | 41.5 | 45.9 | Cross-Entropy ↓ | 2.26 | 2.17 |
| (Stickland et al., 2024) | KL-Then-Steer | System Prompt | Harmfulness | Jailbreaking | Attack Success Rate ↑ | 17.7 | 15.2 | MT-Bench ↑ | 6.43 | 4.44 |
| (Guo et al., 2024) | ASMR | few-shot prompting | Information Retrieval | multi-lingual QA | R@2kt scores ↑ | 52.1 | 50.3 | - | - | - |
| (Qiu et al., 2024) | SEA | ICL | Truthfulness Bias | TruthfulQA BBQ | Info*Truth ↑ Accuracy ↑ | 33.7 59.7 | 33.3 52.9 | - | - | - |
| (Leong et al., 2023) | Theirs | SD | Toxicity | RealToxicityPrompts | Toxicity Probability ↓ | 62.5 | 65.5 | Perplexity | 13.8 | 23.3 |
| (Luo et al., 2024) | PACE | Prompting | Toxicity Faithfulness Sentiment | Jailbreak Generate Biographies Unbiased Generations | Safety Score ↑ Faithfulness Score ↑ 70.5 | 84.2 46.3 58.6 | 61.8 45.1 | MMLU ↑ MMLU ↑ | 45.6 45.8 | 33.8 34.4 |
| (Cao et al., 2024b) | SCANS | Prompting | Harmfulness | Reducing Overrefusal | Correct Refusal Rate ↑ | 97.9 | 89.4 | - | - | - |
| (Wang et al., 2024b) | InferAligner | Goal Prompt | Harmfulness | Jailbreaks | Attack Success Rate ↓ | 0.1 | 14.6 | Accuracy | 58.2 | 58.5 |
| (Liu et al., 2024a) | CtrlA | FLARE | Honesty | RAG | Accuracy ↑ | 69.1 | 60.4 | - | - | - |
| (Li et al., 2024e) | I2CL | ICL | Performance | In-Context Learning | Accuracy ↑ | 75.1 | 76.37 | - | - | - |
| (Wang et al., 2024c) | ACT | Few-shot prompting | Truthfulness | TruthfulQA | True*Info ↑ | 42.3 | 39.5 | - | - | - |
| (Li et al., 2024a) | DAT | GCG | Harmfulness | multi-turn jailbreak | Attack Success Rate ↑ | 23.9 | 25.5 | - | - | - |
| (Liu et al., 2024b) | ICV | ICL | Formality Sentiment | Informal → Formal Translation Negative → Positive Translation | Formality ↑ Positivity ↑ | 48.3 75.3 | 33.0 63.4 | ROUGE-1 ↑ ROUGE-1 ↑ | 80.2 68.3 | 83.9 73.9 |

# D  Papers that compare RepE to Prompting, Fine-tuning and Decoding-based methods

To allow for reproducibility of our meta-study in Section 11.2, we list the papers from which we gathered the data about experimental comparisons between RepE and prompting, fine-tuning and decoding-based methods.

**Prompting.** Zhang et al. (2024c); Cai et al. (2024); Konen et al. (2024); Arditi et al. (2024); Ghandeharioun et al. (2024); Zhang & Viteri (2024); Chen et al. (2024b); Stickland et al. (2024); Guo et al. (2024); Qiu et al. (2024); Leong et al. (2023); Li et al. (2024c); Luo et al. (2024); Cao et al. (2024b); Scalena et al. (2024); Wang et al. (2024b); Liu et al. (2024b); Li et al. (2024e); Liu et al. (2024a); Wang et al. (2024c); Li et al. (2024a)

**Fine-tuning.** Lucchetti & Guha (2024); Li et al. (2024d); Yang et al. (2024); Price et al. (2024); Pham & Nguyen (2024); Liu et al. (2024b); Wang et al. (2024b); Wu et al. (2024b); Yu et al. (2024); Leong et al. (2023); Qiu et al. (2024); Guo et al. (2024); Stickland et al. (2024); Qian et al. (2024); Chen et al. (2024b); Zhang et al. (2024b); Liu et al. (2023); Wang et al. (2024a); Ackerman (2024); Li et al. (2024b); Yin et al. (2024); Cao et al. (2024a); Wu et al. (2024a;c)

**Decoding-based methods.** Li et al. (2024a;d); Cao et al. (2024b); Banerjee et al. (2024); Leong et al. (2023); Qiu et al. (2024); Zhang et al. (2024b)

**Combination.** Wang et al. (2024c); Li et al. (2023a); Stickland et al. (2024); Chen et al. (2024b)

## D.1  Table of empirical comparisons

Furthermore, we list the empirical comparisons between RepE and prompting and fine-tuning methods in Table 14 and Table 15 respectively. Here we include the steering effect and, if available, a measure for model quality for each concept that was compared. When scores for multiple models or subsets of the test dataset are given we average them into one number. Lastly, these table do not include all papers listed above, since we only select papers that report numbers and leave out papers that solely show performance on a graph. This is because reading values of a graph can be unprecise.

Table 15: Collection of studies that compare the steering effect and model quality between RepE and Fine-tuning methods

| Paper | RepE method | Fine-tuning method | Concept | Evaluation Task | Steering Effect | | | Model Quality | | |
|---|---|---|---|---|---|---|---|---|---|---|
| | | | | | Metric | RepE | Fine-tuning | Metric | RepE | Fine-tuning |
| (Li et al., 2024d) | DESTEIN | DISCUP | Toxicity | RealToxicityPrompts | Toxicity Probability ↓ | 0.20 | 0.30 | Perplexity ↓ | 37.8 | 51.9 |
| (Yang et al., 2024) | | SFT | Semantic Consistency | QA | Standard deviation of accuracy ↓ | 3.14 | 2.01 | Accuracy ↑ | 68.7 | 79.65 |
| (Chen et al., 2024b) | TrFr | SFT | Truthfulness | truthful QA | True*Info ↑ | 41.5 | 36.1 | ↓ | 2.26 | 2.1 |
| (Stickland et al., 2024) | KL-Then-Steer | LoRA-DPO | Harmfulness | Jailbreaking | Attack Success Rate ↑ | 17.7 | 14.3 | MT-Bench ↑ | 6.43 | 6.43 |
| (Guo et al., 2024) | ASMR | few-shot prompting | Information Retrieval | multi-lingual QA | R@2kt scores ↑ | 52.1 | 48.7 | - | - | - |
| (Qiu et al., 2024) | SEA | LoRA-FT | Truthfulness | TruthfulQA | Info*Truth ↑ | 33.7 | 42.4 | - | - | - |
| (Leong et al., 2023) | Theirs | DAPT | Toxicity | RealToxicityPrompts | Toxicity Probability ↓ | 62.5 | 57.0 | Perplexity ↓ | 13.8 | 22.47 |
| (Wang et al., 2024b) | InferAligner | SFT | Harfmulness | Jailbreaks | Attack Success Rate ↓ | 0.1 | 9.3 | Accuracy | 58.2 | 56.6 |
| (Pham & Nguyen, 2024) | HPR | LoRA | Truthfulness | TruthfulQA | MC1 ↑ | 53.1 | 40.8 | - | - | - |
| (Liu et al., 2024b) | ICV | ICL | Formality / Sentiment | Informal → Formal Translation / Negative → Positive Translation | Formality ↑ / Positivity ↑ | 48.3 / 75.3 | 22.0 / 63.4 | ROUGE-1 ↑ / ROUGE-1 ↑ | 80.2 / 68.3 | 80.1 / 66.9 |
| (Wu et al., 2024b) | APNEAP | DP | Privacy | Private data leakage | Risk ↓ | 48.4 | 51.1 | Valid-PPL ↓ | 9.3 | 11.4 |
| (Yu et al., 2024) | ReFAT | CAT | Harmfulness | Jailbreaks | Attack Success Rate ↓ | 11.7 | 21.2 | MMLU ↑ | 58.6 | 57.8 |
| (Ackerman, 2024) | Rep Tuning | SFT | Truthfulness | Truthful QA | % Truthful ↑ | 62.5 | 56.5 | - | - | - |
| (Zhang et al., 2024b) | TruthX | SFT | Truthfulness | TruthfulQA | True*Info ↑ | 65.5 | 36.1 | - | - | - |
| (Liu et al., 2023) | RAHF | DPO | Human Preferences | Instruction Following | Win % ↑ | 87.4 | 83.7 | Accuracy ↑ | 57.3 | 56.5 |
| (Wang et al., 2024a) | Model Surgery | DPO / LoRA-FT / LoRA-FT | Toxicity / Refusal / Positivity | Reduce Toxicity / Prevent Jailbreaks / Attitude Adjustement | % Toxicity ↓ / Refusal Rate ↑ / % Positive ↑ | 39.9 / 77.4 / 54.8 | 68.7 / 73.7 / 56.8 | Accuracy ↑ | 34.9 | 35.6 |
| (Yin et al., 2024) | LoFiT | LoRA | Performance | Truthful & Multi-hop QA | Avg Accuracy ↑ | 75.2 | 74.9 | QA Accuracy ↑ | 60.5 | 58.8 |
| (Yin et al., 2024) | BiPO | DPO LoRA | Power-Seeking | Persona Adjustement | Behavior Score ↑ | 2.8 | 2.1 | - | - | - |
| (Wu et al., 2024a) | RED | LoRA-FT | Performance | Accuracy ↑ | Avg score ↑ | 84.3 | 94.7 | - | - | - |
| (Cao et al., 2024b) | SCANS | Prompting | Harmfulness | Reducing Overrefusal | Correct Refusal Rate ↑ | 97.9 | 90.3 | - | - | - |
| (Qian et al., 2024) | Steering Vector | Full-FT | Truthfulness | TruthfulQA | Truth*Info ↑ | 0.68 | 0.41 | Avg Accuracy ↑↑ | 0.39 | 0.31 |
| (Li et al., 2024b) | RMU | LLMU | Dangerous Knowledge | Unlearning | Accuracy ↓ | 29.7 | 49.5 | MT-Bench ↑ | 7.1 | 1.0 |
| (Cai et al., 2024) | SelfControl | DPO | Human Preferences | Alignment | WinRate ↑ | 52.2 | 60.0 | - | - | - |

