# OpenReview forum: "Taxonomy, Opportunities, and Challenges of Representation Engineering for Large Language Models"
_TMLR — Accepted by TMLR_

### Review · Reviewer_WPjs · 2025-04-02

**Summary Of Contributions:**

### Summary of Contributions

This paper presents the first comprehensive survey of Representation Engineering (RepE) for Large Language Models (LLMs), a novel paradigm for controlling model behavior by directly manipulating internal representations. The authors:

1. **Unify existing approaches** under a framework that describes RepE as a pipeline comprising representation identification, operationalization, and control.
2. **Systematically categorize** and compare various RepE methods, highlighting their differences in concept identification, operationalization, and control mechanisms.
3. **Document applications** of RepE across domains like AI safety, ethics, and interpretability, showcasing its versatility.
4. **Compare RepE to traditional methods** (prompting, fine-tuning) through a meta-survey, demonstrating its advantages in effectiveness and efficiency.
5. **Identify challenges and opportunities**, providing a roadmap for future research to improve RepE's reliability, scalability, and applicability.

The paper synthesizes insights from over 130 papers, making it a valuable resource for researchers and practitioners interested in model interpretability and control.

**Audience:**

Yes

**Broader Impact Concerns:**

### Broader Impact Concerns

The paper would benefit from a more detailed **Broader Impact Statement** addressing:
- **Ethical implications**: The potential for RepE to be used to manipulate model behavior in harmful ways (e.g., generating biased or deceptive content).
- **Dual-use concerns**: The risk of RepE being exploited to bypass safety measures in LLMs.
- **Transparency and accountability**: How RepE can be used to enhance model transparency versus how it might obscure model behavior if misused.

**Claims And Evidence:**

Yes

**Requested Changes:**

### Requested Changes

**Critical Changes:**
1. **Add reproducibility details**: Include a section or appendix with practical guidelines for implementing RepE methods, such as code snippets or pseudocode.
2. **Deepen critical analysis**: Expand the discussion on limitations and potential failures of RepE methods, providing concrete examples where they may not work as expected.

**Optional Strengthening Changes:**
1. **Benchmarking section**: Add a more detailed analysis of existing benchmarks and evaluation metrics for RepE, possibly including a table comparing them.
2. **Case studies**: Include a few detailed case studies demonstrating the application of RepE in real-world scenarios to illustrate its practical utility.
3. **Broader impact statement**: Address ethical implications more thoroughly, such as potential misuse of RepE for malicious purposes.

**Strengths And Weaknesses:**

### Strengths and Weaknesses

**Strengths:**
- **Comprehensive coverage**: The survey is thorough, covering all major aspects of RepE from foundational concepts to advanced applications.
- **Clear taxonomy**: The proposed framework effectively categorizes diverse RepE methods, making the field more accessible.
- **Empirical comparisons**: The meta-survey comparing RepE to other methods provides concrete evidence of its advantages.
- **Practical insights**: The discussion of challenges and opportunities offers actionable guidance for future research.


**Weaknesses:**
- **Lack of reproducibility details**: While the paper reviews many methods, it does not provide enough detail on how to reproduce key results or implement these techniques.
- **Limited critical analysis**: The survey could benefit from a deeper critique of the limitations and potential pitfalls of RepE methods.
- **Incomplete coverage of benchmarks**: The discussion on evaluating RepE methods is somewhat superficial; more detail on existing benchmarks and evaluation metrics would be helpful.

---

> ### Author Response · Authors · 2025-07-02
>
> We thank the reviewer for their valuable feedback.
>
> Reproducibility Details: We believe the results we generated as part of the survey are reproducible by following our literature survey methodology described in Appendix B. Since our contribution does not include experiments or algorithms, we do not include any code or pseudocode.
> We believe including pseudocode of specific RepE methods would not greatly improve reproducibility while requiring a lot of work from us. Readers who want to reproduce methods or want to inspect their code are best served by looking at the papers that propose the method (which we always cite) and their respective codebases. Further, there are ~100 RepE methods, and including the pseudocode of many of them would be considerable effort, esp since not all of them provide pseudocode.
>
> Critical Analysis: We believe we provide a thorough discussion of limitations and pitfalls in Section 12. As per your suggestion, we will extend this by better illustrating these weaknesses and challenges by giving examples of failures. Concretely, we will try to add a short description of real or hypothetical scenarios where such a weakness could lead to failures to the points in Section 12.
>
> Benchmarking Section: We agree that the survey would be strengthened with a discussion of Evaluation Methodologies. Thus, we are now working on adding a section that describes existing metrics and methodologies for evaluating RepE methods, suggests best practices for carrying out evaluations and discusses the two existing RepE benchmarks while making suggestions for a future benchmark.
>
> Broader Impact Statement: We are happy to add a short discussion of the dual risk concerns of RepE methods.

---

> > ### Author Response · Authors · 2025-07-07
> >
> > Based on the reviewers' suggestions, we have made multiple changes:
> >
> > 1. We have added concrete examples of failures, each weakness and limitations in Section 13 and deepened our discussion about assumptions in feature geometries in Section 13.2.2.
> > 2. We have added a full section discussing Evaluation Methodologies. It explains and categorises existing methodologies for evaluation RepE and describes existing benchmarks. Furthermore, we give concrete best practices to follow when evaluating RepE methods and guidelines on how to design a benchmark.
> > 3. We have added a Broader Impact Statement discussing potential downsides of RepE.

---

### Review · Reviewer_FgAo · 2025-06-13

**Summary Of Contributions:**

This paper focuses on representation engineering (RepE) for large language models (LLMs), which focus on manipulating the internal representations of a model to control its behavior with respect to specific concepts. It systematically introduces three key components of this process:
- Representation Identification: how to identify and characterize the relationship between internal representations and human-interpretable concepts.
- Representation Operationalization: how to formalize these relationships in a usable form.
- Representation Control: how to steer the model’s behavior by applying a steering function to its activations or weights.

At last, the paper presents practical methods and pipelines for implementing these techniques, along with further insights and considerations for effective application.

**Audience:**

Yes

**Broader Impact Concerns:**

The paper raises potential concerns regarding the ethical implications and safety risks of RepE, particularly in its application for controlling LLM behavior at a fine-grained level. These concerns are appropriately acknowledged and discussed in Sections 9.1 (AI Safety) and 9.2 (Ethics), where the authors highlight both the opportunities and the risks of RepE, such as misuse, stealth interventions, or value misalignment.

Given that the paper explicitly dicusses these issues and provides mitigation strategies and research needs, I do not have additional broader impact concerns beyond what has been presented.

**Claims And Evidence:**

Yes

**Requested Changes:**

1. Clarify and expand Section 2 for general LLM audiences: Consider providing more introductory detail in Section 2 when introducing the concept of RepE for LLMs. The current presentation uses terminology that may only be accessible to readers already familiar with the RepE literature. To improve clarity and accessibility:
    - Provide concrete examples of how existing work applies RepE to LLMs or to other applications.

    - Briefly outline major strategies or categories used in prior RepE work and RepE pipeline.

   - Articulate the ultimate goals and motivations of RepE in LLMs.

These additions would make the section more informative for newcomers with LLM background.

2. Expand Appendix D into a more structured literature review:
The paper references many prior works that compare RepE with prompting, fine-tuning, and decoding-based methods, to highlight the potential of RepE. Consider expanding Appendix D into a brief literature review or summary table that captures the task, setting, approach of each paper. This would help readers form a clearer picture of how RepE area progresses and performs in practice and strengthen the survey’s contribution as a reference resource.

**Strengths And Weaknesses:**

### Strengths

1. Timely and impactful contribution: This paper provides the first comprehensive survey on Representation Engineering (RepE) for LLMs—a timely and increasingly important topic given the limitations of prompting and post-training methods. In light of the recent surge of interest in RepE, this work fills a critical gap in the LLM research landscape.

2. Clear structure and systematic analysis: The paper is well-structured and easy to follow. It systematically introduces how RepE can benefit LLMs, summarizes existing methods, identifies key bottlenecks, and outlines opportunities for future research. The proposed pipeline offers a coherent structure for categorizing a wide range of techniques. The inclusion of formal notation, illustrative figures, and detailed comparisons adds further clarity.

3. Forward-looking insights and actionable directions: Beyond being descriptive, the paper provides a thoughtful discussion of practical challenges (e.g., multi-concept steering, robustness) and promising research directions (e.g., benchmarking standards, non-linear geometries). These forward-looking insights elevate the paper from a literature summary to a roadmap for future work in RepE.


### Weaknesses

1. Limited empirical synthesis and discussion depth: While the meta-survey spans over a hundred papers, it lacks a quantitative synthesis or deeper comparative analysis of method effectiveness. The practical value would be significantly enhanced by summarizing trade-offs (e.g., control effectiveness vs. degradation, data requirements, computational cost) or highlighting differences across use cases. For instance, Section 10 and Appendix D list papers that claim RepE outperforms other methods, but offer little discussion on how these results were obtained. As it stands, the reader is left with paper counts rather than meaningful insights.

2. Insufficient critique of underlying assumptions: Many RepE methods are built on strong assumptions about the geometry of representations (e.g., linear directions in activation space). However, the paper does not sufficiently interrogate the validity of these assumptions or the consequences when they fail. A more rigorous empirical or theoretical critique would help clarify the generalizability of existing methods.

---

> ### Author Response · Authors · 2025-07-02
>
> We thank the reviewer for their valuable feedback. We’d like to address some of the mentioned weaknesses and ask for clarity on the requested changes.
>
> Weaknesses:
> We agree it would be useful to offer a stronger empirical synthesis comparing different methods. We attempted our best to gather this information, but found that there is not much empirical evidence that would allow us to make thorough comparisons of RepE methods. All the evidence we were able to gather is summarised in Section 7.2 as best practices. We acknowledge this is a limitation of our work and will highlight this as future work.
>
> We critique the underlying assumptions of RepE in section 12.2. For the mentioned assumption “linear directions in activation space” we attempt to highlight the assumption and its impacts in Section 12.2.2, discuss its validity in Section 5.1 and suggest opportunities for going beyond those assumptions in Section 13.2. To make it easier for the reader, we will add references between these parts of the paper.
>
> Requested Changes:
> We agree it is important to make Section 2 understandable to readers not yet familiar with the RepE literature. To clarify Section 2 for a general audience, we will:
> - introduce a running example of an application of RepE
> - Give a short summary of different strategies in Section 2
> - We will add a discussion of the goals of RepE to 2.2
>
> We are happy to add a table in Appendix D that summarises the concepts, tasks and experimental outcomes of papers that combined RepE with other methods.

---

> > ### Author Response · Authors · 2025-07-07
> >
> > Based on the suggestions by the reviewer, we have made three changes:
> >
> > 1. We have attempted to make Section 2 more accessible to readers who are new to RepE by introducing examples of how RepE can be used, describing the goals of RepE, and outlining some perspectives that help to grasp RepE and describe major strategies within RepE.
> > 2. We have gathered the results of empirical studies where RepE is compared to prompting and fine-tuning in Tables 14 and 15.
> > 3. We have deepened the discussion about "assumptions about feature geometries" in 13.2.2 to describe how assumptions could fall short and the consequences when they fail.
> >
> > We believe this addresses the requested changes by the reviewer.

---

### Review · Reviewer_ug2U · 2025-06-24

**Summary Of Contributions:**

This paper provides the first comprehensive survey of Representation Engineering , a growing subfield focused on controlling LLMs by directly manipulating their internal representations. It introduces a unifying framework for categorizing RepE methods into three core steps: representation identification, operationalization, and control. The authors conduct a thorough review of over 130 papers, identify major taxonomic dimensions, compare RepE to related techniques. I think the survey is well structured and also comprehensive of the majority of the literature on this topic at the moment. It requires some minor modifications.

**Audience:**

Yes

**Claims And Evidence:**

Yes

**Requested Changes:**

- In Section 10 (Comparison to Other Methods). Some readers may still confuse RepE with prompt tuning or knowledge editing. Maybe add comparison \ table contrasting RepE with prompt engineering, fine-tuning, and memory editing.
- On Evaluation Metrics and Benchmarks. The authors say that there is no benchmark, so how are current methods evaluated, and according to which metrics ? This should be stated more clearly.
- I don't know if it is possible, but a running example would make the manuscript easier to follow, as in some parts the reader might get lost without a reference example to think of.

**Strengths And Weaknesses:**

**Strengths**
- The field of RepE has rapidly expanded without a clear systematization. This paper offers a much-needed taxonomy and conceptual grounding for the community.
- Covers numerous papers with a clear inclusion criterion (Appendix B) and provides extensive meta-analysis, including a taxonomy of methods, usage statistics, and comparisons with other behavioral control approaches.
- The authors’ proposed three-part pipeline (identification → operationalization → control) is intuitive

**Weaknesses**: see requested changes

---

> ### Author Response · Authors · 2025-07-02
>
> We thank the reviewer for their valuable feedback.
>
> In Section 10, we already discussed the differences of RepE to prompting, model editing and fine-tuning. However, to make it more clear, we will attempt to summarise the differences into a short table.
>
> We agree that the survey would be strengthened with a discussion of Evaluation Methodologies. We are working on adding a section that describes existing metrics and methodologies for evaluating RepE methods, suggests best practices for carrying out evaluations and discusses the two existing RepE benchmarks while making suggestions for a future benchmark.
>
> We will introduce a short running example in Section 2 about applying RepE to make an LLM more honest and will refer back to it in Sections 3-6 as well as when discussing limitations.

---

> > ### Author Response · Authors · 2025-07-07
> >
> > Based on the suggestions of the reviewers, we have made three changes:
> >
> > 1. We have added Table 12, which summarises the key differences of multiple methods to RepE. We think this will make it easier for readers to grasp the difference between RepE and other methods.
> > 2. We have added a full section discussing Evaluation Methodologies. It explains and categorises existing methodologies for evaluation RepE and describes existing benchmarks. Furthermore, we give concrete best practices to follow when evaluating RepE methods and guidelines on how to design a benchmark.
> > 3. We have added a running example of steering a model to be more truthful in Section 2. By recalling this running example in Sections 3-6 and 13, we hope this makes the sections easier to follow.
> >
> > We believe this addresses all suggestions made by the reviewers.

---

### Comment · Editors_In_Chief · 2025-10-10

On October 10, 2025, by request of the authors, the EiCs replaced the camera ready version of this paper. The acknowledgments were inadvertently omitted from the original camera ready version (as well as one broken reference), and are now included.

---

### Decision · Action_Editor_aaV1 · 2025-08-14

**Recommendation:** Accept as is

**Additional Comments:**

This is a strong survey paper with four key areas of strength.
1. **Comprehensive coverage**: The paper analyzes over 130 papers on RepE, which covers the majority of literature on the field to-date.
2. **Clear, well-structured taxonomy**: The proposed framework for understanding RepE methods, which comprises representation identification, operationalization, and control, offers a effective categorization of RepE. The analysis is systematic and well-structured, covering a wide range of related discussion including challenges and opportunities.
3. **Timeliness and impact**: RepE is an increasingly important topic given the limitations of alternatives such as prompting and post-training methods. Reviewers agreed that this paper fills an important gap by providing conceptual grounding and organization to a rapidly growing area.
4. **Quality and depth of insight**: The survey offers a multitude of practical insights and actionable directions for future work. It is valuable not only as an entry point for newcomers but also as a roadmap for future research on RepE. Challenges and opportunities are thoughtfully discussed and there is plenty of insight to be gleaned by reading this work.

During the review phase, some relatively minor concerns were raised by the reviewers. These concerns were resolved by the authors during the rebuttal phase by: (a) adding an extended discussion on evaluation methodologies including best practices and guidelines for designing new benchmarks, (b) improving clarity of the relationship to alternate methods such as prompt engineering and fine-tuning, and (c) improving readability and providing intuition for newcomers to the area.

After the changes, the reviewers' concerns were resolved. In light of this and the strengths highlighted above, I recommend this paper for acceptance as-is. Given the paper’s breadth of coverage and depth of insight, I further recommend survey certification.

**Audience:**

Yes

**Audience Explanation:**

There is a clear audience for this work. Reviewers agreed that RepE is a timely topic and that this paper provides a solid conceptual grounding for this growing area as well as a clearly-structured taxonomy to understand current methods.

**Claims And Evidence:**

Yes

**Claims Explanation:**

This paper surveys the burgeoning subfield of representation engineering (RepE) for LLMs. RepE is distinguished from alternatives such as fine-tuning and prompting by its focus on direct manipulation of the model’s internal representations. This paper introduces a unifying framework for RepE consisting of (a) representation identification, (b) operationalization, and (c) control. It provides a meta-analysis of over 130 papers on RepE along these dimensions. It also summarizes a number of important aspects of RepE, ranging from effective pipelines, evaluation benchmarks, application domains, challenges, and opportunities for future work. The paper delivers on its promise as a comprehensive survey of the area of representation engineering.